# On the Spectral Unreachability of Brain Graph Learning

**Jiaming Zhuo**[1]  **Shuai Zhai**[1]  **Ziyi Ma**[1]  **Kun Fu**[1]  **Chuan Wang**[2]  **Di Jin**[3]  **Zhen Wang**[4]
**Xiaochun Cao**[5]  **Huazhu Fu**[6]  **Liang Yang**[1][*]

## Abstract

Brain network classification is pivotal for diagnosing neurological disorders, yet identifying interpretable functional biomarkers fundamentally relies on precise parcellation. Unfortunately, conventional deep graph encoders applied to brain networks suffer from a critical theoretical limitation termed *Spectral Unreachability*. Through graph spectral analysis, this paper demonstrates that standard coupled encoder-pooling architectures inevitably oversmooth node representations, corrupting the high-frequency topological signals strictly required to delineate sharp module boundaries. To provide a structural remedy, the Hierarchical Spectral Parcellation Network (HiSP-Net) is proposed, which structurally decouples partition learning from feature smoothing via a *project-then-align* paradigm. Specifically, HiSP-Net maps representations directly into a partition space using a topology-agnostic projection block to preserve all-frequency details, while a Topology-Aware Alignment regularizer subsequently enforces spatial coherence. Extensive evaluations demonstrate that HiSP-Net consistently outperforms state-of-the-art baselines in classification, while successfully extracting stable and structurally coherent functional biomarkers. Source code is available at https://github.com/Kevin-916/HiSP-Net-demo/.

[1]Hebei Province Key Laboratory of Big Data Calculation, School of Artificial Intelligence, Hebei University of Technology, Tianjin, China [2]School of Computer Science and Technology, Beijing JiaoTong University, Beijing, China [3]Tianjin Key Laboratory of Cognitive Computing and Application, College of Intelligence and Computing, Tianjin University, Tianjin, China [4]School of Artificial Intelligence, OPtics and ElectroNics (iOPEN), School of Cybersecurity, Northwestern Polytechnical University, Xi'an, China [5]School of Cyber Science and Technology, Shenzhen Campus of Sun Yat-sen University, Shenzhen, China [6]Institute of High Performance Computing, Agency for Science, Technology and Research, Singapore. Correspondence to: Liang Yang <yangliang@vip.qq.com>.

*Proceedings of the $43^{rd}$ International Conference on Machine Learning*, Seoul, South Korea. PMLR 306, 2026. Copyright 2026 by the author(s).

## 1. Introduction

Diagnosing neurological disorders via brain network classification has emerged as a pivotal paradigm (Bullmore & Sporns, 2009). Beyond solely predicting diagnostic labels, a critical imperative for clinical deployment is identifying discriminative biomarkers for conditions such as Alzheimer's disease (Jack Jr et al., 2024) and Autism (Li et al., 2021). However, distinct from generic graph-level classification (Zhang et al., 2018), extracting such interpretable biomarkers fundamentally hinges on functional parcellation, namely, the precise decomposition of the brain into distinct, spatially coherent communities (*i.e.*, functional modules) (Sporns & Betzel, 2016). Neuroscientific evidence indicates that cognitive functions are supported by these modular structures, and their topological disruption is often the pathology hallmark (Eickhoff et al., 2018). Therefore, deriving robust diagnostic features necessitates preserving boundary sharpness without compromising spatial coherence, as blurred boundaries or fragmented regions inevitably dilute the disease-specific connectivity signals required for early detection.

To address this brain graph classification task, existing deep learning models typically adopt a coupled *encoder-pooling* architecture. Within this pipeline, a backbone encoder, operating via either local (*e.g.*, Graph Neural Networks (GNNs) (Li et al., 2021; Cui et al., 2022; Kan et al., 2022a; Zhang et al., 2023)) or global (*e.g.*, Graph Transformers (GTs) (Kan et al., 2022b; Xu et al., 2024; Yu et al., 2024; Peng et al., 2025)) mechanisms, first transforms raw connectivity into latent representations, which are subsequently condensed by a pooling operation. Depending on the coarsening strategy, these models generally fall into three categories: (1) global readout, which applies global aggregation functions directly to the encoder's output to obtain graph-level embeddings (Cui et al., 2022); (2) node selection, which retains a subset of salient nodes based on encoder-derived scores while discarding the rest (Li et al., 2021); and (3) cluster pooling, which aggregates nodes into super-nodes via soft assignment matrices learned from the latent features (Ying et al., 2018). Crucially, in this coupled paradigm, the optimization of pooling partitions is intrinsically constrained and severely biased by the representation limits of the preceding encoder.

This inherent coupling introduces a fundamental flaw: stan-

dard encoders inevitably corrupt the high-frequency topological signals strictly required for precise parcellation (Nt & Maehara, 2019; Dong et al., 2021). Specifically, within the graph spectral domain, identifying distinct functional modules necessitates preserving sharp, step-like spatial transitions (Shuman et al., 2013), which are distinct from the temporal high-frequency physiological noises often encountered in raw BOLD time series (Birn, 2012). On one hand, local GNNs fundamentally act as low-pass filters (Nt & Maehara, 2019), smoothing node features across local neighborhoods to suppress noise. This forced smoothing inevitably erodes the sharp spatial boundaries between functional modules, causing distinct communities to merge indistinguishably (Conflict 1 in Sec. 3.1). On the other hand, global GTs, driven by global attention mechanisms, tend to induce representation homogenization and rank collapse (Dong et al., 2021). This global smoothing dilutes local topological heterogeneity, rendering the node representations indistinguishable (Conflict 2 in Sec. 3.1). Thus, the downstream pooling operation is compelled to operate on these compromised representations, *i.e.*, formally established as Spectral Unreachability (Theorem 1). Driven by such coupling, existing models generally yield partitions that are either spatially fragmented or spectrally oversmoothed, failing to reconstruct structurally coherent functional brain organization.

To resolve this limitation, a *project-then-align* paradigm is proposed, fundamentally decoupling partition learning from representation smoothing. Realizing this paradigm, the Hierarchical Spectral Parcellation Network (HISP-NET) is introduced as a structural remedy. Departing from conventional smoothing-based encoders, HISP-NET is constructed by stacking Spectral Parcellation layers. In the project phase, a node-wise MLP serves as a spectrally unconstrained backbone to directly map node representations into a partition assignment matrix. By bypassing the representation smoothing inherent in GNNs and GTs, this design effectively preserves high-frequency spectral components, retaining the sharp boundaries essential for partitioning. Then, the align phase imposes structural regularization via the Topology-Aware Alignment (TAA) regularizer. This constraint explicitly enforces spatial coherence, preventing the fragmentation typical of unconstrained topology-agnostic methods. By recursively applying these decoupled layers, HISP-NET establishes a hierarchy capable of capturing multi-scale functional organizations. Theoretically, HISP-NET is proven to preserve gradient norms independent of depth, thereby circumventing spectral unreachability and guaranteeing the reconstruction of stable partitions with both boundary sharpness and structural coherence.

The main contribution of this paper can be summarized as

- We discover and formally identify *Spectral Unreachability* as a critical theoretical bottleneck in coupled encoder-pooling architectures, which fundamentally limits the learning of sharp module boundaries.

- We propose HISP-NET, a novel hierarchical framework that adopts a *project-then-align* paradigm to structurally decouple partition learning from representation smoothing, ensuring both gradient preservation and spatial coherence.

- Extensive evaluations on benchmark datasets demonstrate that HISP-NET consistently outperforms state-of-the-art baselines in brain disease diagnosis, while successfully extracting stable and structurally coherent functional biomarkers.

## 2. Preliminaries

This section first presents the brain network classification problem and outlines spectral geometry concepts. Then, it reviews existing graph encoders and pooling paradigms.

### 2.1. Problem Definition and Notations.

This study focuses on brain network classification for neurological disease diagnosis. Let $\mathcal{D} = \{(G_i, y_i)\}_{i=1}^{M}$ denote a dataset comprising $M$ subjects, where $G_i \in \mathcal{G}$ represents the brain graph of the $i$-th subject and $y_i \in \mathcal{Y} = \{1, \ldots, C\}$ denotes the categorical label (*e.g.*, Patient vs. Control). The objective is to learn a mapping function $f : \mathcal{G} \rightarrow \mathcal{Y}$. Formally, each brain network is defined as an attributed graph $G_i = (\mathcal{V}, \mathcal{E}, \mathbf{X})$ containing $N$ Regions of Interest (ROIs). $\mathcal{V}$ denotes the set of anatomical nodes ($|\mathcal{V}| = N$). The connectivity is encoded by a weighted adjacency matrix $\mathbf{A} \in \mathbb{R}^{N \times N}$, where each entry $a_{ij}$ quantifies the functional connectivity strength (*e.g.*, correlation magnitude) between ROI $i$ and $j$. Typically, the connectivity profile of each node is utilized as its intrinsic attribute; thus, the node attribute matrix is set as $\mathbf{X} \in \mathbb{R}^{N \times D}$, where the dimension $D = N$.

### 2.2. Spectral Geometry on Graphs

To analyze the topological smoothness of the defined node attributes, the normalized Laplacian matrix is introduced as $\mathbf{L} = \mathbf{I} - \mathbf{D}^{-1/2} \mathbf{A} \mathbf{D}^{-1/2}$, where $\mathbf{D}$ denotes the diagonal degree matrix with entries $d_{ii} = \sum_j a_{ij}$. From a graph signal processing perspective (Ortega et al., 2018), the variation of a signal $\mathbf{X}$ across an edge $(i, j)$ is quantified by the local graph gradient: $(\nabla_{\mathcal{G}} \mathbf{x})_{ij} = \sqrt{a_{ij}}(\frac{\mathbf{x}_j}{\sqrt{d_{jj}}} - \frac{\mathbf{x}_i}{\sqrt{d_{ii}}})$. Thus, the global smoothness is measured by the Dirichlet Energy:

$$\begin{aligned} \mathcal{E}_{Dir}(\mathbf{X}) &= \mathrm{Tr}(\mathbf{X}^\top \mathbf{L} \mathbf{X}) \\ &= \frac{1}{2} \sum_{(i,j) \in \mathcal{E}} a_{ij} \| \frac{\mathbf{x}_i}{\sqrt{d_{ii}}} - \frac{\mathbf{x}_j}{\sqrt{d_{jj}}} \|^2. \end{aligned} \quad (1)$$

Minimizing $\mathcal{E}_{Dir}$ enforces consistency among connected nodes, effectively suppressing high-frequency components

(*i.e.*, smoothing sharp boundaries) (Nt & Maehara, 2019). Note that the high-frequency components herein strictly refer to the graph Laplacian's upper spectrum (*i.e.*, sharp spatial boundaries), fundamentally distinct from temporal physiological noises in BOLD fMRI.

## 2.3. Graph Encoders

**Graph Neural Networks (GNNs).** Standard GNNs, *e.g.*, GCN (Jiang et al., 2019), typically update node representations via recursive message passing (Zhuo et al., 2023; 2024a;c;b; Yang et al., 2025a; He et al., 2025; Shan et al., 2026; Wang et al., 2026), serving as the backbone for brain network analysis models. A layer is formulated as:

$$\mathbf{H}^{(l+1)} = \sigma(\hat{\mathbf{A}}\mathbf{H}^{(l)}\mathbf{W}^{(l)}), \tag{2}$$

where $\hat{\mathbf{A}}$ denotes the Laplacian-normalized adjacency matrix and $\mathbf{W}$ represents the learnable network parameter. Although implemented in the spatial domain, this operation is theoretically equivalent to a spectral filtering process (Kipf, 2016). Specifically, left-multiplying by $\hat{\mathbf{A}}$ approximates a first-order polynomial filter $g(\lambda) \approx 1 - \lambda$ on the Laplacian eigenvalues (Wu et al., 2019). Thus, stacking such layers functions as a *low-pass filter*, exponentially attenuating the high-frequency components (large $\lambda$) required for sharp boundary delineation (Nt & Maehara, 2019).

**Graph Transformers (GTs).** To capture long-range dependencies, GTs employ global attention mechanisms to update node representations (Zhuo et al., 2025a;b). That is,

$$\mathbf{H}^{(l+1)} = \text{Softmax}\left(\mathbf{Q}\mathbf{K}^{\top}/\sqrt{D}\right)\mathbf{V}, \tag{3}$$

where $\mathbf{Q} = \mathbf{H}^{(l)}\mathbf{W}_Q$, $\mathbf{K} = \mathbf{H}^{(l)}\mathbf{W}_K$, and $\mathbf{V} = \mathbf{H}^{(l)}\mathbf{W}_V$. Distinct from local GNNs, the attention mechanism acts as a *global mixer*. It generates a dense connectivity matrix, effectively treating the brain network as an implicit fully connected graph. By globally aggregating features, GTs function as extreme low-pass filters that induce representation homogenization and rank collapse (Dong et al., 2021), thereby eradicating the explicit local topological boundaries required for fine-grained partitioning.

## 2.4. Graph Pooling on Brain Networks

In graph classification tasks, pooling operations are essential to reduce dimensionality and extract a holistic graph-level representation from node representations $\mathbf{H} \in \mathbb{R}^{N \times D}$. Existing pooling strategies in brain network analysis fall into three primary paradigms, namely,

- **Global Pooling (Readout).** Standard baselines (*e.g.*, BrainGB (Cui et al., 2022)) compress graphs via flat operations like Mean/Max, which completely discards topological hierarchies and precludes the generation of node-level parcellations.

- **Node Selection Pooling (Dropping).** ROI-selection models like BRAINGNN (Li et al., 2021) retain only salient biomarkers. While identifying key regions, this strategy results in incomplete brain maps, making it unsuitable for dense functional partitioning.

- **Node Clustering Pooling (Grouping).** Hierarchical mapping models (*e.g.*, adapting DIFFPOOL (Ying et al., 2018)) map all $N$ nodes to $K$ clusters via a soft assignment $\mathbf{S}$ ($K \ll N$), ensuring whole-brain coverage.

This study focuses on the node clustering paradigm, which aligns mathematically with the goal of functional parcellation in neuroscience (Eickhoff et al., 2018). Formally, this task aims to learn a cluster assignment matrix $\mathbf{C} \in \mathbb{R}^{N \times K}$, where $K$ is the number of functional modules. In spectral graph theory, the optimal partition $\mathbf{C}^*$ should maximize the standard Normalized Cut objective (Shi & Malik, 2000):

$$\mathcal{J}_{Ncut}(\mathbf{C}) = \frac{\text{Tr}(\mathbf{C}^{\top}\mathbf{A}\mathbf{C})}{\text{Tr}(\mathbf{C}^{\top}\mathbf{D}\mathbf{C})}, \tag{4}$$
$$\text{s.t. } \mathbf{C} \in \{0,1\}^{N \times K}, \ \mathbf{C}\mathbf{1}_K = \mathbf{1}_N.$$

Although Eq. 4 serves as the theoretical objective function, maximizing it directly poses an NP-hard problem.

## 3. Methodology

This section introduces the proposed Hierarchical Spectral Parcellation Network (HISP-NET) for brain network analysis. First, motivated by the theoretical limitations of existing deep graph encoders, the underlying spectral conflict is analyzed. Next, the specific architecture and its core modules are detailed as a structural remedy. Finally, a comprehensive model analysis is given.

### 3.1. Analysis: The Spectral Unreachability

In this section, the spectral limitations of deep graph learning frameworks used in brain network analysis (*e.g.*, GNNs and GTs reviewed in Sec. 2.3) are analyzed in the context of functional partitioning. Here, partitioning is formulated as a *latent structure discovery process* (Eickhoff et al., 2018), mapping noisy functional connectivity (Birn, 2012) (analogous to noisy edges in graph structure learning (Jin et al., 2020; Zhu et al., 2021)) into coherent functional modules.

Recalling the graph signal processing perspective mentioned in Sec. 2.2, distinguishing distinct functional modules necessitates a partition matrix $\mathbf{C}$ with *sharp boundaries* (*i.e.*, step-like transitions containing significant high-frequency components (Shuman et al., 2013)).

**The Theoretical Conflict: Smoothness vs. Sharpness.**

Ideally, the search for the optimal partition matrix $\mathbf{C}^*$ should cover the entire space of valid assignments to maximize the

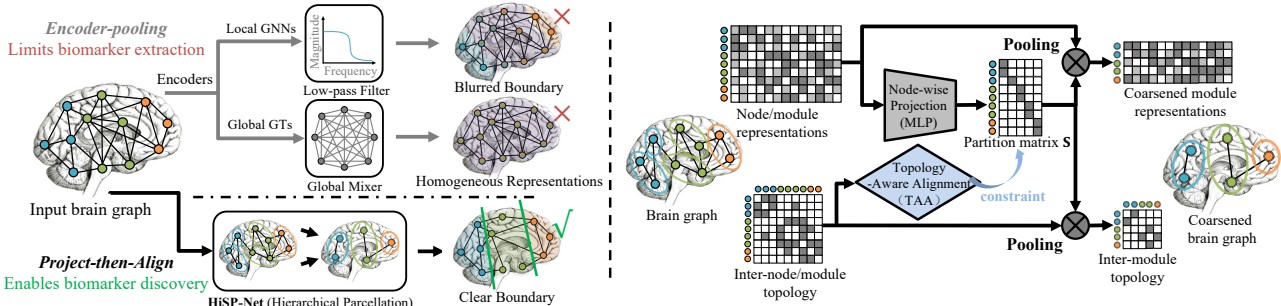

(a) Architecture Comparison in Brain Analysis        (b) The proposed Spectral Parcellation block

*Figure 1.* Illustration of the motivation and the proposed HISP-NET. (a) Architecture Comparison: Existing coupled encoder-pooling models (Top) suffer from spectral unreachability (established in Theorem 1), acting as low-pass filters that blur functional boundaries. In contrast, the proposed *project-then-align* paradigm (Bottom) decouples partition learning, preserving all-frequency signals to enable the extraction of stable and structurally coherent biomarkers. (b) A detailed view of the Spectral Parcellation block. Specifically, it employs a topology-agnostic projection (*i.e.*, a node-wise MLP) to map representations directly into a partition space, explicitly avoiding topological smoothing during partition generation. This is constrained by a Topology-Aware Alignment (TAA) regularizer to enforce spatial coherence, followed by disentangled coarsening to reconstruct multi-scale functional organizations.

objective defined in Eq. 4. However, standard paradigms restrict the search for $\mathbf{C}$ to a constrained hypothesis space $\mathcal{H}_\Theta$ dictated by the topological smoothing of the backbone architecture. This constrained optimization is formulated as

$$\max_{\mathbf{C}} \mathcal{J}_{Ncut}(\mathbf{C}) \ \ \text{s.t.} \ \mathbf{C} \in \mathcal{H}_\Theta = \{\Psi_\theta(\mathbf{A}, \mathbf{X}) | \theta \in \Theta\}, \quad (5)$$

where $\Psi_\theta(\cdot)$ denotes the parameterized mapping function of the backbone model (*e.g.*, GNNs) with learnable parameters $\theta$. The central paradox lies in the *spectral mismatch* between the objective (Eq. 4) and the architectural constraints ($\mathcal{H}_\Theta$). A sharp partition $\mathbf{C}^*$ inherently contains significant energy in the *high-frequency band* of the graph spectrum (Shuman et al., 2013). In contrast, standard backbones impose constraints that suppress these critical frequencies, namely:

- **GNN Conflict (Forced Smoothing):** As detailed in Sec. 2.3, GNN propagation approximates a low-pass filter $g(\lambda) \approx (1 - \lambda)^k$. Given that the functional connectivity inherently contains noisy inter-module edges ($a_{ij} > 0$), this filtering mechanism forces the representations of connected (yet functionally distinct) nodes to become similar. This explicitly penalizes the high local variation (defined in Sec. 2.2) required to define sharp boundaries, resulting in over-smoothed partitions.

- **GT Conflict (Global Homogenization):** The attention mechanism (Eq. 3) in GTs acts as a global mixer, effectively creating a fully connected graph. This dense global connectivity acts as an extreme low-pass filter, where the distinct local features of a node are diluted by the influx of global information. This inevitably induces representation homogenization and rank collapse, obscuring the fine-grained high-frequency variations necessary for sharp partitioning.

**Theorem 1** (Spectral Unreachability). *Let $\mathbf{C}^*$ be a ground-truth sharp partition matrix. Its spectral decomposition*

*contains significant energy in the high-frequency band of the graph spectrum. However, standard backbones (GNNs and GTs) function as low-pass operators, confining their output to a smoothed hypothesis space $\mathcal{H}_{smooth}$. Thus, there exists a theoretical lower bound on the approximation error:*

$$\min_{\mathbf{C} \in \mathcal{H}_{smooth}} \|\mathbf{C} - \mathbf{C}^*\|_F \geq \|\mathcal{P}_{high}(\mathbf{C}^*)\|_F = \epsilon > 0, \quad (6)$$

*where $\mathcal{P}_{high}$ denotes the projection onto the high-frequency spectral components that are intrinsically suppressed by the backbone's architectural constraints.*

Theorem 1 indicates that while $\mathbf{C}^*$ necessitates high Dirichlet energy (due to sharp boundaries), both GNN propagation and GT attention act as energy-minimizing operators, filtering out the necessary high-frequency components. Refer to Appendix B for the full proof.

*Remark* 1 (The Limitation of Implicit Modeling). Empirical studies on BQN (Yang et al., 2025b) demonstrate that utilizing universal approximators (MLPs) yields superior performance, theoretically by circumventing the spectral unreachability (Theorem 1). However, functioning as end-to-end implicit learners, such architectures map inputs to diagnosis without resolving the partition matrix $\mathbf{C}$. This opacity not only forfeits structured denoising (*i.e.*, pruning noisy edges) but also sacrifices mechanistic interpretability, rendering them incapable of extracting the structurally coherent biomarkers required for clinical deployment.

### 3.2. HISP-NET

Based on the theoretical analysis in the previous subsection, this subsection introduces the Hierarchical Spectral Parcellation Network (HISP-NET). To resolve the aforementioned Spectral Conflict, HISP-NET fundamentally departs from the coupled encoder-pooling paradigm. Instead, HISP-NET

adopts a *project-then-align* paradigm: performing partitioning in a topology-agnostic manner before enforcing structural consistency. As shown in Fig. 1, HISP-NET utilizes a spectrally unconstrained backbone to project nodes into a high-dimensional latent space, followed by an objective-driven alignment to recover graph topology. This ensures the hypothesis space covers the full spectral bandwidth required for sharp partitioning, ultimately enabling the discovery of interpretable functional biomarkers.

**Overview.** HISP-NET consists of $L$ stacked spectral parcellation layers, designed to progressively transform node-level signals into multi-scale functional module representations. Formally, let $\mathcal{G}^{(l)} = (\mathbf{H}^{(l)}, \mathbf{A}^{(l)})$ denote the brain graph at layer $l$, where $\mathbf{H}^{(l)} \in \mathbb{R}^{N_l \times D}$ represents the node representation matrix, $\mathbf{A}^{(l)} \in \mathbb{R}^{N_l \times N_l}$ denotes the node topology, and $N_l$ is the number of nodes at layer $l$. The input layer receives the initial brain graph, *i.e.*, $\mathbf{H}^{(0)} = \mathbf{X}$ and $\mathbf{A}^{(0)} = \mathbf{A}$. Each hierarchical layer $l$ performs partition learning and graph coarsening through three strictly decoupled phases: (1) Spectral Projection: A multi-head projection block that maps node representations $\mathbf{H}^{(l)}$ to latent cluster assignments, serving as the spectrally unconstrained basis. (2) Topology-Aware Alignment: A structural regularizer that explicitly injects topological inductive biases into the partitions via auxiliary optimization objectives. (3) Disentangled Coarsening: A differentiable pooling operation that utilizes the aligned partitions to aggregate representations, producing the coarsened graph $\mathcal{G}^{(l+1)}$ for the subsequent layer.

### 3.2.1. SPECTRAL PROJECTION

The spectral projection block aims to generate a spectrally unconstrained soft partition matrix $\mathbf{S}^{(l)}$ (acting as a continuous relaxation of the ideal discrete $\mathbf{C}^*$) and a structurally denoised topology $\tilde{\mathbf{A}}^{(l)}$ from the input $\mathbf{H}^{(l)}$ and $\mathbf{A}^{(l)}$. This is achieved through a three-stage workflow: (1) Initial Projection, (2) Dual-Stream Ensemble, and (3) Gated Fusion.

**Initial Projection.** To bypass the representation smoothing, a topology-agnostic projection is adopted. A node-wise MLP directly projects the node representations $\mathbf{H}^{(l)}$ into a latent cluster space. This can be formulated as

$$\mathbf{U}^{(l)} = \text{MLP}(\mathbf{H}^{(l)}) \in \mathbb{R}^{N_l \times N_{l+1}}, \quad (7)$$

where $N_l > N_{l+1}$. Acting as a spectrally unconstrained operator, it preserves the full-spectrum information inevitably attenuated by standard smoothing constraints (Sec. 2.3).

**Decoupled Representation-Topology Ensemble.** To independently mitigate noise in node representations and graph topology, we adopt a decoupled dual-stream ensemble.

(1) *Representation Stream:* The latent partition logits $\mathbf{U}^{(l)}$ are projected into $M$ parallel subspaces, aiming to model diverse functional patterns. The projection is expressed as a channel-wise affine transformation, *i.e.*,

$$\mathbf{Z}^{(l,m)} = \mathbf{U}^{(l)} \odot \boldsymbol{\alpha}_m + \boldsymbol{\beta}_m, \ m \in \{1, \dots, M\}, \quad (8)$$

where $\boldsymbol{\alpha}_m$ and $\boldsymbol{\beta}_m \in \mathbb{R}^{1 \times N_{l+1}}$ are broadcasted parameters.

(2) *Topology Stream:* To reduce the impact of unreliable connections, $M$ stochastic topological views are generated via Bernoulli masking (Rong et al., 2020). For each head $m$, a perturbed topology $\mathbf{A}^{(l,m)}$ is sampled as:

$$a_{ij}^{(l,m)} \sim a_{ij}^{(l)} \cdot \text{Bernoulli}(1-p), \quad (9)$$

where $p$ is a perturbation rate. This creates a structurally diverse ensemble, preventing the architecture from overfitting to a single fixed topology containing spurious correlations.

**Dual-Stream Gated Fusion.** To synthesize the two streams, a context-aware gating network is employed. Conditioned on the global graph representation $\mathbf{c}_{global}^{(l)} = \text{Mean}(\mathbf{H}^{(l)}) \in \mathbb{R}^{D_l}$, a reliability vector $\mathbf{w}^{(l)} \in \mathbb{R}^M$ is learned via

$$\mathbf{w}^{(l)} = \text{Softmax}(f_{gate}(\mathbf{c}_{global}^{(l)})), \quad (10)$$

where $f_{gate}(\cdot)$ denotes the gating network (*e.g.*, an MLP).

These weights perform a soft selection over the ensemble, adaptively amplifying informative subspaces while suppressing noisy views. Then, the final partition $\mathbf{S}^{(l)}$ and consensus topology $\tilde{\mathbf{A}}^{(l)}$ for layer $l$ are derived as

$$\mathbf{S}^{(l)} = \text{Softmax}\Big( \sum_{m=1}^{M} w_m^{(l)} \mathbf{Z}^{(l,m)}/\tau \Big), \quad (11)$$

$$\hat{\mathbf{A}}^{(l)} = \sum_{m=1}^{M} w_m^{(l)} \mathbf{A}^{(l,m)}, \quad (12)$$

where $\tau$ stands for a temperature controlling the sharpness of the partition distribution. To accommodate the negative correlations inherent in fMRI data for downstream tasks, the absolute consensus topology is defined as $\tilde{\mathbf{A}}^{(l)} = |\hat{\mathbf{A}}^{(l)}|$. This fusion allows $\mathbf{S}^{(l)}$ to focus on informative semantics and $\tilde{\mathbf{A}}^{(l)}$ to serve as a structurally denoised basis for subsequent alignment and coarsening.

### 3.2.2. TOPOLOGY-AWARE ALIGNMENT (TAA)

To regularize the spectrally unconstrained soft partition $\mathbf{S}^{(l)}$, structural inductive biases are injected via a joint auxiliary objective. This comprises two terms: a differentiable MinCut loss, which serves as an equivalent objective to maximizing $\mathcal{J}_{Ncut}$, and a Frobenius-normalized orthogonality constraint. The TAA loss at layer $l$ is expressed as:

$$\mathcal{L}_{taa}^{(l)} = -\frac{\text{Tr}(\mathbf{S}^{(l)\top} \tilde{\mathbf{A}}^{(l)} \mathbf{S}^{(l)})}{\text{Tr}(\mathbf{S}^{(l)\top} \tilde{\mathbf{D}}^{(l)} \mathbf{S}^{(l)})}$$
$$+ \left\| \frac{\mathbf{S}^{(l)\top} \mathbf{S}^{(l)}}{\|\mathbf{S}^{(l)\top} \mathbf{S}^{(l)}\|_F} - \frac{\mathbf{I}_{N_{l+1}}}{\sqrt{N_{l+1}}} \right\|_F, \quad (13)$$

where $\tilde{\mathbf{D}}^{(l)}$ denotes the diagonal degree matrix of $\tilde{\mathbf{A}}^{(l)}$, and $\|\cdot\|_F$ is the Frobenius norm. Specifically, the first term acts as an intra-module cohesive force to group tightly connected regions, while the second term functions as an inter-module repulsive force to prevent trivial collapsed partitions. Their synergy guarantees the structural coherence of the final parcellation. Crucially, utilizing the absolute consensus topology provides a necessary mathematical relaxation for signed networks. It prevents zero-degree instability from negative correlations and guarantees the positive semi-definiteness required for valid MinCut optimization.

### 3.2.3. DISENTANGLED COARSENING

Leveraging the aligned partition matrix $\mathbf{S}^{(l)}$ and the structurally denoised topology $\tilde{\mathbf{A}}^{(l)}$, differentiable spectral pooling is performed to aggregate node-level information into functional module representations for the next layer.

**Differentiable Pooling.** Both node representations and topology are spectrally coarsened from $N_l$ nodes to $N_{l+1}$ functional modules using the learned assignment $\mathbf{S}^{(l)}$, *i.e.*,

$$
\begin{aligned}
\mathbf{H}^{(l+1)} &= \mathbf{S}^{(l)\top}\mathbf{H}^{(l)} \ \in \mathbb{R}^{N_{l+1}\times D}, \\
\mathbf{A}^{(l+1)} &= \mathbf{S}^{(l)\top}\tilde{\mathbf{A}}^{(l)}\mathbf{S}^{(l)} \ \in \mathbb{R}^{N_{l+1}\times N_{l+1}},
\end{aligned}
\tag{14}
$$

where $\mathbf{H}^{(l+1)}$ encodes the aggregated representations of functional modules and $\mathbf{A}^{(l+1)}$ captures the inter-module connectivity. The coarsened graph $\mathcal{G}^{(l+1)}$ then serves as input for the next hierarchical layer.

**Final Readout.** After $L$ layers of hierarchical coarsening, the final representation $\mathbf{H}^{(L)}$ is flattened and projected via a prediction head to generate the diagnostic brain fingerprint:

$$
\mathbf{h}_{final} = f_{out}(\mathrm{vec}(\mathbf{H}^{(L)})),
\tag{15}
$$

where $\mathrm{vec}(\cdot)$ stands for the flattening operation (concatenating rows into a vector), and $f_{out}(\cdot)$ represents the learnable projection head (implemented as an MLP).

**Objective Functions.** The overall objective integrates diagnostic task supervision (*i.e.*, cross-entropy loss $\mathcal{L}_{ce}$) with the joint structural regularization summed across all layers. The architecture is trained end-to-end via

$$
\mathcal{L}_{total} = \mathcal{L}_{ce}(\hat{\mathbf{y}}, \mathbf{y}) + \lambda \sum_{l=0}^{L-1} \mathcal{L}_{taa}^{(l)},
\tag{16}
$$

where $\hat{\mathbf{y}} = \mathrm{Softmax}(\mathbf{h}_{final})$ denotes the predicted probability distribution, and $\lambda$ is a trade-off scalar controlling the strength of the structural constraints.

### 3.3. Model Analysis

Compared to existing brain representation learning models (predominantly built upon GNNs and GTs), HISP-NET

offers distinct theoretical advantages regarding gradient dynamics and inductive bias alignment.

**Avoiding Pooling Collapse via Gradient Preservation.** The quality of graph pooling hinges on the discriminative power of node representations. Standard architectures inherently act as smoothing operators that suppress high-frequency components: GNNs smooth representations locally via Laplacian smoothing, while GTs enforce global homogenization via dense attention. This creates a fundamental conflict for parcellation: both mechanisms lead to representation oversmoothing, causing trivial partitions (*i.e.*, the assignment matrix $\mathbf{S}^{(l)}$ fails to distinguish functional modules). In contrast, HISP-NET employs a node-wise MLP, which is structurally decoupled from the graph topology and thus avoids intrinsic representation smoothing.

**Proposition 2.** *Unlike GNNs and GTs, where the gradient norm vanishes with depth (due to inherent spectral filtering or global homogenization), the* HISP-NET *representation admits a gradient norm that is asymptotically independent of the architectural constraints, allowing the preservation of arbitrary high-frequency components.*

**Decoupling Architecture from Regularization.** Standard paradigms suffer from the spectral conflict (Sec. 3.1), where architectural constraints restrict the hypothesis space $\mathcal{H}_\Theta$ to fixed or low-frequency subspaces. Specifically, this coupling mandates forced smoothing (GNNs) or global homogenization (GTs). In contrast, HISP-NET utilizes the spectrally unconstrained spectral projection block (Sec. 3.2.1) to decouple representation learning from topological constraints. This shifts the role of topological regularizer entirely to the proposed objective, ensuring that smoothing is driven strictly by optimization needs rather than being an unavoidable artifact of the backbone.

## 4. Experiments

This section conducts comprehensive evaluations on three real-world datasets (ABIDE, ADHD-200, and ADNI) to verify the effectiveness and interpretability of the proposed HISP-NET. The evaluations aim to (1) test the performance of diagnosing neurological diseases against state-of-the-art *Graph Neural Networks (GNNs)*-based models, including: GCN (Kipf, 2016), GAT (Veličković et al., 2017), DIFF-POOL (Ying et al., 2018), BRAINGNN (Li et al., 2021), BRAINGB (Cui et al., 2022), FBNETGEN (Kan et al., 2022a), and A-GCL (Zhang et al., 2023), *Graph Transformers (GTs)*-based models, including SAN (Kreuzer et al., 2021), GRAPHORMER (Ying et al., 2021), GRAPHTRANS (Wu et al., 2021), BRAINNETTF (Kan et al., 2022b), CONTRASTPOOL (Xu et al., 2024), ALTER (Yu et al., 2024), and BIOBGT (Peng et al., 2025), and a *Neural Network (NN)*-based model, *i.e.*, BQN (Yang et al., 2025b); (2) verify the capability of HISP-NET in identifying biologically

*Table 1.* Performance comparison of the proposed HiSP-NET against 15 state-of-the-art baselines across three benchmark datasets. Results are reported as mean$_{\pm\text{std}}$ over five runs. **Bold** and underlined indicate the best and second-best results, respectively.

| | Method | ABIDE | | ADHD-200 | | ADNI | | |
|---|---|---|---|---|---|---|---|---|
| | | AUC ↑ | ACC ↑ | AUC ↑ | ACC ↑ | AUC ↑ | ACC ↑ | RANK ↓ |
| *GNNs* | GCN | $59.59_{\pm3.44}$ | $59.30_{\pm3.38}$ | $67.01_{\pm3.56}$ | $64.92_{\pm6.32}$ | $62.45_{\pm3.63}$ | $59.12_{\pm4.53}$ | 12.50 |
| | GAT | $60.43_{\pm3.88}$ | $60.10_{\pm4.13}$ | $61.97_{\pm3.28}$ | $63.38_{\pm3.18}$ | $62.00_{\pm2.88}$ | $58.75_{\pm2.86}$ | 12.83 |
| | DIFFPOOL | $62.34_{\pm3.42}$ | $56.12_{\pm4.61}$ | $67.64_{\pm4.08}$ | $63.85_{\pm2.87}$ | $66.21_{\pm4.83}$ | $63.80_{\pm4.64}$ | 11.00 |
| | BRAINGNN | $64.42_{\pm3.57}$ | $63.09_{\pm1.35}$ | $67.19_{\pm2.86}$ | $65.16_{\pm3.81}$ | $61.81_{\pm1.58}$ | $58.72_{\pm3.14}$ | 11.50 |
| | BRAINGB | $70.32_{\pm3.66}$ | $65.12_{\pm3.90}$ | $75.23_{\pm11.02}$ | $69.34_{\pm7.41}$ | $66.44_{\pm3.33}$ | $63.70_{\pm4.65}$ | 7.67 |
| | FBNETGEN | $74.55_{\pm3.77}$ | $67.09_{\pm3.37}$ | $77.40_{\pm4.76}$ | $68.82_{\pm6.27}$ | $67.05_{\pm2.16}$ | $63.26_{\pm1.38}$ | 7.00 |
| | A-GCL | $73.86_{\pm2.91}$ | $71.04_{\pm2.40}$ | $74.78_{\pm4.39}$ | $73.11_{\pm4.30}$ | $68.15_{\pm3.33}$ | $67.08_{\pm4.57}$ | 5.00 |
| *GTs* | SAN | $71.35_{\pm2.18}$ | $65.34_{\pm2.91}$ | $51.22_{\pm2.21}$ | $51.09_{\pm2.00}$ | $66.11_{\pm3.41}$ | $61.78_{\pm4.22}$ | 11.67 |
| | GRAPHORMER | $63.91_{\pm4.05}$ | $61.88_{\pm6.85}$ | $58.64_{\pm1.50}$ | $61.60_{\pm0.90}$ | $60.69_{\pm5.26}$ | $55.75_{\pm3.18}$ | 13.67 |
| | GRAPHTRANS | $60.13_{\pm6.73}$ | $57.83_{\pm4.71}$ | $51.49_{\pm1.15}$ | $50.76_{\pm2.07}$ | $61.41_{\pm3.65}$ | $58.60_{\pm5.41}$ | 14.83 |
| | BRAINNETTF | $77.93_{\pm1.41}$ | $69.26_{\pm2.26}$ | $79.79_{\pm3.14}$ | $72.67_{\pm3.17}$ | $69.73_{\pm2.61}$ | $67.85_{\pm2.92}$ | 4.33 |
| | CONTRASTPOOL | $57.36_{\pm0.87}$ | $57.44_{\pm0.69}$ | $71.19_{\pm2.26}$ | $69.16_{\pm2.85}$ | $68.17_{\pm3.28}$ | $66.21_{\pm3.90}$ | 9.83 |
| | ALTER | $77.99_{\pm2.21}$ | $70.10_{\pm2.26}$ | $83.16_{\pm1.61}$ | $73.48_{\pm1.38}$ | $71.86_{\pm3.13}$ | $66.92_{\pm3.93}$ | 3.50 |
| | BIOBGT | $69.96_{\pm1.18}$ | $69.70_{\pm2.92}$ | $71.64_{\pm1.14}$ | $71.06_{\pm0.08}$ | $63.16_{\pm3.74}$ | $62.27_{\pm3.23}$ | 8.17 |
| *NN* | BQN | $\mathbf{79.85_{\pm1.27}}$ | $\underline{72.53_{\pm1.41}}$ | $\underline{83.34_{\pm1.13}}$ | $\underline{75.68_{\pm1.95}}$ | $\underline{74.18_{\pm3.34}}$ | $\underline{68.62_{\pm3.22}}$ | $\underline{1.83}$ |
| | HiSP-NET (OURS) | $\underline{78.05_{\pm1.55}}$ | $\mathbf{72.90_{\pm1.32}}$ | $\mathbf{83.72_{\pm1.17}}$ | $\mathbf{76.51_{\pm1.82}}$ | $\mathbf{78.54_{\pm2.40}}$ | $\mathbf{77.99_{\pm3.04}}$ | **1.17** |

*Figure 2.* Comparison of 64-cluster parcellation results produced by the proposed HiSP-NET against DIFFPOOL and K-MEANS for NC and ASD groups, and their difference maps on the ABIDE dataset. L and R indicate the left and right hemispheres, respectively.

meaningful biomarkers; and (3) validate the necessity of the components via ablation studies. Detailed introduction of the datasets, baseline models, and implementation settings, and additional results are provided in the Appendix A.1.

### 4.1. Disease Diagnosis Performance

Tab. 1 reports the quantitative comparison between the proposed HiSP-NET and 15 baseline models across three benchmark datasets. Overall, HiSP-NET shows consistent superiority over state-of-the-art baselines. From this table, three key observations can be summarized: (1) HiSP-NET maintains the highest average rank, achieving best or runner-up performance across all metrics. This consistency indicates that the proposed design effectively generalizes across

varying disease patterns, whereas baselines exhibit significant performance fluctuations. (2) On the ADNI dataset (limited sample size), HiSP-NET outperforms the runner-up by a substantial margin of 9.37% in ACC. This improvement highlights the model's data efficiency, suggesting that the proposed decoupled learning strategy is particularly advantageous in small-sample scenarios where conventional models often falter. (3) HiSP-NET consistently surpasses coupled encoder-pooling models (GNNs and GTs) across all datasets. This performance gap empirically demonstrates the superiority of the *project-then-align* paradigm over smoothing-based models in distinguishing complex brain network patterns.

**Generalization and Robustness Evaluation.** Beyond standard binary classification on single-site data, HiSP-NET

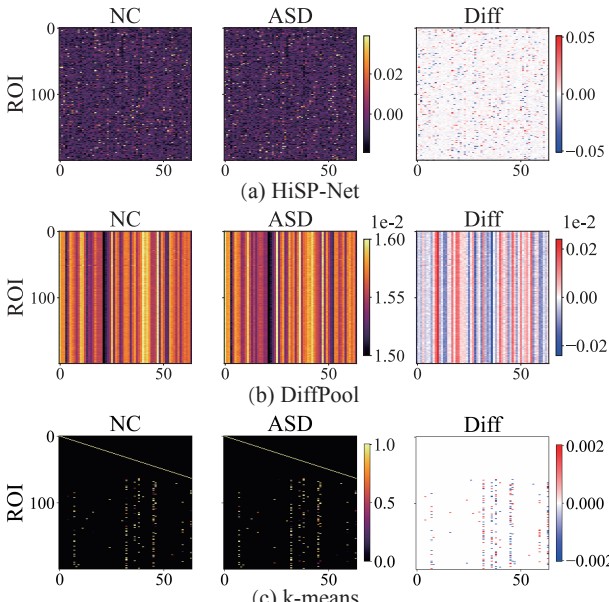

*Figure 3.* Heatmap visualization of clustering assignments ($K = 64$) for NC, ASD, and difference groups on the ABIDE dataset.

was further evaluated under highly challenging clinical scenarios: multi-class disease subtyping (ADNI (6-class) and PPMI) and cross-site Out-of-Distribution (OOD) settings. The empirical evaluations reveal that HISP-NET consistently outperforms specialized baselines (*e.g.*, BRAINOOD) across these heterogeneous conditions. These results conclusively demonstrate its superior generalization capability and robustness against clinical domain shifts. Detailed experimental configurations and comprehensive quantitative results are given in Sections A.2.1 and A.2.2, respectively.

### 4.2. Interpretability and Qualitative Analysis

To verify whether the proposed HISP-NET yields biologically meaningful and discriminative partitions, the clustering results ($K = 64$) on the ABIDE dataset are visualized. The proposed HISP-NET is benchmarked against two representative models: K-MEANS (Ahmed et al., 2020), which serves as a topology-agnostic baseline, and DIFFPOOL, a coupled encoder-pooling baseline. The comparative results are illustrated on the cortical surface (Fig. 2) and via ROI-assignment heatmaps (Fig. 3).

**Spatial Coherence vs. Mode Collapse.** Fig. 2 illustrates that HISP-NET achieves an optimal balance between spatial coherence and local granularity. Specifically, DIFFPOOL in Fig. 2(b) exhibits severe mode collapse, with the cortex dominated by only ∼3 oversized clusters (visualized as large, uniform color blocks). This confirms that coupled GNN-based pooling tends to excessively suppress high-frequency topological signals. Moreover, K-MEANS in Fig. 2(c) generates overly fragmented partitions (resembling salt-and-pepper noise) due to the absence of structural constraints.

In contrast, HISP-NET in Fig. 2(a) produces spatially contiguous and structurally coherent functional modules. The generated boundaries are sharp yet distinct, validating that the topology-agnostic projection constrained by TAA (in Sec. 3.2.2) effectively preserves topological information.

Fig. 3 provides the statistical explanation for the patterns observed above by comparing the ROI-to-cluster assignment probabilities and their signed difference ($NC - ASD$). For the coupled baseline DIFFPOOL in Fig. 3(b) exhibits a repetitive vertical bar-code pattern, indicating an input-invariant state where dominant clusters are assigned globally regardless of ROI location. Thus, its difference map shows only systematic, column-wise shifts rather than localizable ROI-specific deviations. For the topology-agnostic K-MEANS in Fig. 3(c) displays a sharp diagonal pattern, suggesting a trivial solution where clusters merely mimic ROI indices. Its difference map is sparse and unstructured, failing to detect meaningful group variations. The proposed HISP-NET in Fig. 3(a) reveals heterogeneous and structured discrepancies. Instead of global shifts, distinct element-wise variations (red/blue regions) are observed. These pronounced shifts indicate that the proposed project-then-align paradigm successfully captures disease-specific functional reorganizations, aligning with recognized biomarkers in the DMN and VIS regions (Benkarim et al., 2021; Padmanabhan et al., 2017; Lombardo et al., 2019).

To quantitatively validate the structural stability of the identified biomarkers, the Adjusted Rand Index (ARI) of the generated parcellations was evaluated across five independent trials. HISP-NET achieves a high ARI of $0.89 \pm 0.05$, outperforming the coupled baseline DIFFPOOL (with an ARI of $0.83 \pm 0.05$). This confirms that the distinct functional regions discovered by the project-then-align paradigm are highly robust and reproducible.

### 4.3. Ablation Study on Pooling Architecture

Fig. 4 summarizes the performance of HISP-NET under various pooling configurations to validate the proposed architecture. The results primarily underscore the necessity of hierarchical pooling. The baseline using direct global readout (labeled as "-") consistently yields the lowest scores, whereas incorporating pooling leads to significant improvements (*e.g.*, $+6.57\%$ AUC on ADHD-200), confirming that coarse-grained structural representations are essential for diagnosis. Furthermore, hierarchical configurations (*e.g.*, $[128, 32]$) generally outperform single-layer counterparts (*e.g.*, $[128]$), validating the benefit of multi-scale parcellation in capturing both local and global patterns. However, extending the depth to three layers (e.g., $[128, 64, 32]$) yields diminishing returns compared to two-layer settings. This indicates that a two-stage pooling strategy strikes the optimal balance between feature abstraction and model complexity.

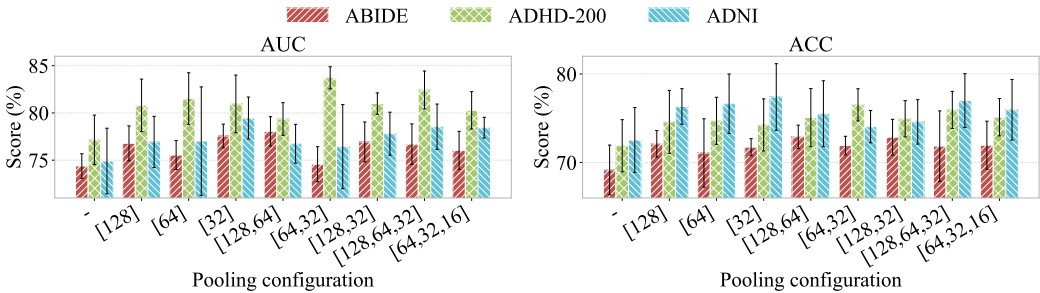

*Figure 4.* Ablation study on the proposed pooling with varying hierarchical configurations across the ABIDE, ADHD-200, and ADNI.

## 5. Conclusions

In this work, spectral unreachability is identified as a fundamental barrier restricting existing graph encoders from learning sharp brain parcellations. To resolve this, HISP-NET is proposed, introducing a *project-then-align* paradigm to structurally decouple representation learning from topological constraints. Theoretical analysis and extensive experiments verify that this design effectively preserves high-frequency spectral components, yielding superior diagnostic performance and clinically interpretable biomarkers. Crucially, HISP-NET bridges the gap between the expressivity of deep learning and the interpretability required by neuroscience. Future research directions include extending this spectral decoupling perspective to dynamic functional connectivity analysis and multi-modal brain networks.

## Acknowledgements

This work was supported in part by the National Natural Science Foundation of China (No. 92570118, U22B2036, 62376088, 62272020, 62025604, 92370111, 62272340, 62261136549, 52441501), in part by the Hebei Natural Science Foundation (No. F2024202047), in part by the National Science Fund for Distinguished Young Scholarship (No. 62025602), in part by the Hebei Yanzhao Golden Platform Talent Gathering Programme Core Talent Project (Education Platform) (HJZD202509), in part by the Post-graduate's Innovation Fund Project of Hebei Province (CXZZBS2025036), in part by the Tencent Foundation, and in part by the XPLORER PRIZE.

## Impact Statement

This paper presents work whose goal is to advance the field of Machine Learning. There are many potential societal consequences of our work, none which we feel must be specifically highlighted here.

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

# A. Experimental Details

## A.1. Experimental Setup

**Datasets.** Four real-world fMRI datasets are adopted for brain network analysis.

- Autism Brain Imaging Data Exchange (**ABIDE**)[1]. This dataset is primarily designed to investigate the differences in functional brain connectivity between individuals with Autism Spectrum Disorder (ASD) and normal controls (NC). The data are obtained by preprocessing resting-state functional magnetic resonance imaging (fMRI) scans collected from 17 international sites. The resulting dataset consists of 516 ASD subjects and 493 NC subjects.

- Attention Deficit Hyperactivity Disorder (**ADHD-200**)[2]. This dataset is a multi-site dataset obtained from 8 international imaging sites for the study of Attention Deficit Hyperactivity Disorder (ADHD). The data used in this paper consist of 459 subjects, including 230 typically developing individuals and 229 ADHD subjects.

- Alzheimer's Disease Neuroimaging Initiative (**ADNI**)[3]. ADNI is one of the most influential and widely used data resources in the fields of neuroscience and aging research. It has been extensively applied to studies of Alzheimer's disease (AD), including fundamental research, machine learning–based diagnostic modeling, neuroimaging analysis, and biomarker discovery. Due to access restrictions on the official ADNI website, the dataset used in this study consists of 71 NC subjects and 53 AD subjects. In addition, a 6-class version is also provided for cross-site experiments, comprising 1327 subjects collected from 59 sites. This dataset includes 819 NC, 73 significant memory concern (SMC), 102 late mild cognitive impairment (LMCI), 179 mild cognitive impairment (MCI), 89 early mild cognitive impairment (EMCI), and 65 AD samples.

- Parkinson's Progression Markers Initiative (**PPMI**). It is a multi-center longitudinal study designed to identify biomarkers associated with the onset and progression of Parkinson's disease. The PPMI dataset used in this study includes 209 subjects, comprising 113 patients with Parkinson's disease (PD), 67 prodromal subjects, 14 individuals with SWEDD (scans without evidence of dopaminergic deficit), and 15 normal controls (NC).

The brain is parcellated into regions of interest (ROIs) using the Craddock 200 functional atlas (Craddock et al., 2012) for ABIDE and ADHD-200, which provides a functionally

homogeneous parcellation of the whole brain. The mean blood-oxygen-level dependent (BOLD) time series within each ROI is extracted to represent regional neural activity. Specifically, 200 ROIs are used for the ABIDE dataset, while 190 ROIs are adopted for the ADHD-200 dataset. For ADNI, the AAL-90 atlas (Tzourio-Mazoyer et al., 2002) is employed, comprising 90 ROIs.

**Baselines.** To comprehensively evaluate the performance of the proposed model, three categories of methods are considered, comprising a suite of state-of-the-art baselines.

- Graph Neural Networks (**GNNs**)-based models. The GNN-based methods include three classical graph neural networks, namely GCN (Kipf, 2016), GAT (Veličković et al., 2017) and DiffPool (Ying et al., 2018), as well as four brain-specific GNN models: BrainGNN (Li et al., 2021), BrainGB (Cui et al., 2022), FBNetGen (Kan et al., 2022a), and A-GCL (Zhang et al., 2023).

- Graph Transformers (**GTs**)-based models. The GT-based methods include three classical graph transformers, namely SAN (Kreuzer et al., 2021), Graphormer (Ying et al., 2021), and GraphTrans (Wu et al., 2021), as well as four brain-specific graph transformer models: BrainNETTF (Kan et al., 2022b), ContrastPool (Xu et al., 2024), ALTER (Yu et al., 2024), and BioBGT (Peng et al., 2025).

- Neural Network (**NN**)-based model. The NN-based methods include one brain-specific model, namely BQN (Yang et al., 2025b).

**Metrics.** To comprehensively evaluate the performance of the proposed model, three widely used evaluation metrics from machine learning and medical diagnosis are adopted, including: (1) Area Under the Receiver Operating Characteristic Curve (AUC), which measures the model's overall ability to distinguish between positive and negative samples and reflects its classification stability; (2) Accuracy (ACC), which evaluates the overall classification correctness; (3) RANK denotes the average ranking of the same model across all datasets and all evaluation metrics.

**Implementation Details.** All experiments are conducted on a Linux server equipped with an NVIDIA GeForce RTX 3090 GPU using PyTorch and PyTorch Geometric (PyG) for training, validation, and testing. All datasets are randomly split into training, validation, and test sets with a ratio of 7:1:2. Each model is trained for 200 epochs, and the final results are reported as the average over 5 random runs. The Adam optimizer is adopted with an initial learning rate of $1 \times 10^{-4}$, a target learning rate of $1 \times 10^{-5}$, and a weight decay of $1 \times 10^{-4}$. The batch size is selected from $\{64, 32, 16\}$, the activation function is set to LeakyReLU, and the dropout rate is chosen from $\{0.1, 0.2, 0.3\}$.

---

[1] http://preprocessed-connectomes-project.org/abide/

[2] https://fcon_1000.projects.nitrc.org/indi/adhd200/

[3] https://adni.loni.usc.edu/

*Table 2.* Performance comparison of baseline models on the multi-class datasets ADNI (6-class) and PPMI (mean ± std). **Bold** and underlined indicate the best and second-best results, respectively.

| Model | ADNI (6-class) | | | PPMI | | |
|---|---|---|---|---|---|---|
| | AUC ↑ | ACC ↑ | F1 ↑ | AUC ↑ | ACC ↑ | F1 ↑ |
| GCN | $57.82 \pm 1.91$ | $55.67 \pm 3.08$ | $24.74 \pm 2.95$ | $57.86 \pm 3.76$ | $51.23 \pm 3.18$ | $24.47 \pm 4.61$ |
| BRAINNETTF | $76.54 \pm 4.39$ | $66.63 \pm 2.56$ | $30.09 \pm 9.12$ | $59.08 \pm 2.19$ | $51.67 \pm 6.24$ | $25.86 \pm 4.92$ |
| BIOBGT | $61.33 \pm 5.98$ | $52.08 \pm 2.08$ | $32.29 \pm 2.31$ | $62.64 \pm 4.10$ | $51.72 \pm 4.26$ | $27.41 \pm 3.18$ |
| ALTER | $77.58 \pm 2.30$ | $66.15 \pm 2.11$ | $32.97 \pm 5.32$ | $66.71 \pm 3.11$ | $\mathbf{52.78 \pm 3.27}$ | $\underline{28.74 \pm 3.54}$ |
| BQN | $78.12 \pm 3.14$ | $68.74 \pm 2.36$ | $33.86 \pm 3.20$ | $67.23 \pm 3.20$ | $50.42 \pm 6.64$ | $26.48 \pm 7.52$ |
| HISP-NET | $\mathbf{79.13 \pm 5.92}$ | $\mathbf{69.18 \pm 2.27}$ | $\mathbf{51.42 \pm 3.26}$ | $\mathbf{68.48 \pm 3.67}$ | $\underline{52.25 \pm 5.69}$ | $\mathbf{39.84 \pm 6.39}$ |

*Table 3.* Performance comparison on ABIDE and ADNI (6-class) datasets under a site-holdout OOD setting (mean ± std). **Bold** and underlined indicate the best and second-best results, respectively.

| Model | ABIDE | | ADNI (6-class) | |
|---|---|---|---|---|
| | AUC | ACC | AUC | ACC |
| GCN | $61.71 \pm 4.59$ | $61.85 \pm 4.39$ | $62.28 \pm 8.95$ | $60.84 \pm 4.05$ |
| GAT | $63.07 \pm 4.67$ | $63.12 \pm 4.72$ | $62.25 \pm 7.54$ | $60.26 \pm 5.12$ |
| BRAINGNN | $59.76 \pm 3.93$ | $60.00 \pm 3.96$ | $63.42 \pm 7.54$ | $62.89 \pm 4.65$ |
| CONTRASTPOOL | $62.57 \pm 3.93$ | $62.09 \pm 3.15$ | $63.75 \pm 2.59$ | $60.34 \pm 5.01$ |
| BRAINOOD | $63.52 \pm 4.28$ | $63.95 \pm 4.65$ | $65.27 \pm 3.78$ | $64.78 \pm 5.26$ |
| HISP-NET | $\mathbf{67.25 \pm 1.41}$ | $\mathbf{66.37 \pm 1.90}$ | $\mathbf{68.83 \pm 1.46}$ | $\mathbf{66.91 \pm 1.86}$ |

## A.2. Additional Experimental Results

### A.2.1. MULTI-CLASS CLASSIFICATION

To evaluate the diagnostic capability beyond standard binary classification, HISP-NET was further evaluated under challenging multi-class clinical scenarios using the multi-site ADNI (6-class disease subtyping) and PPMI (4-class progression tracking) datasets. As reported in Tab. 2, HISP-NET demonstrates substantial superiority, achieving the highest performance across nearly all evaluation metrics on both datasets, with the sole exception of a highly competitive second-best accuracy on the PPMI cohort. These results conclusively indicate that the structurally coherent functional representations learned by HISP-NET possess fine-grained discriminative power, successfully capturing the subtle topological shifts across distinct disease severities to enable precise clinical subtyping.

### A.2.2. CROSS-SITE OOD EVALUATION

To assess model robustness against clinical distribution shifts induced by heterogeneous scanners and protocols, cross-site Out-of-Distribution (OOD) evaluations were conducted on the multi-site ABIDE and ADNI (6-class) datasets, strictly following the benchmark protocol established by BRAINOOD (Xu et al., 2025). As shown in Tab. 3, HISP-NET consistently outperforms both standard deep graph architectures and the specialized OOD baseline, BRAINOOD, across all metrics on both multi-center cohorts. This superior generalization confirms that the topology-agnostic *project-then-align* paradigm successfully prevents

the model from overfitting to site-specific spurious correlations, thereby extracting invariant and clinically robust neuroimaging biomarkers.

### A.2.3. ABLATION STUDY ON TAA COMPONENTS

Tab. 4 isolates the individual contributions of the MinCut term $\mathcal{L}_{mc}$ and the orthogonality constraint $\mathcal{L}_{ortho}$ on the ABIDE dataset. First and foremost, it can be concluded that the removal of either component triggers a clear performance degradation. Specifically, excluding $\mathcal{L}_{mc}$ yields an accuracy drop of $-3.11\%$ ACC, indicating its necessity in guiding node aggregation via topological connectivity. Eliminating $\mathcal{L}_{ortho}$ separately leads to a $-3.39\%$ decrease in ACC. Notably, these single-term variants underperform even the fully unconstrained baseline (*i.e.*, w/o $\mathcal{L}_{mc}$ & $\mathcal{L}_{ortho}$), which maintains the second-best accuracy of $71.61\%$. This performance gap points to potential optimization imbalances induced by isolated constraints, such as trivial cluster collapse or maximum-entropy feature dilution. The joint TAA loss counteracts these individual bottlenecks by balancing intra-module cohesion with inter-module diversity, thereby stabilizing the underlying parcellation space.

### A.2.4. PARAMETER SENSITIVITY ANALYSIS

**Impact of Pooling Heads (Fig. 5).** The proposed HISP-NET demonstrates remarkable stability across different head configurations. On smaller datasets (ABIDE and ADNI), performance fluctuations are minimal, indicating that HISP-NET is not overly sensitive to head redundancy. In contrast, on the larger ADHD-200 dataset, performance peaks at 8 heads. This suggests that while a single head suffices for smaller datasets, moderate multi-head aggregation ($H = 8$) effectively enhances the representation capacity required for capturing complex heterogeneity in complex datasets.

**Impact of Pooling Loss Ratio (Fig. 6).** This evaluation tracks the robustness of the TAA regularizer against varying scales of the hyperparameter $\lambda$. As illustrated in Fig. 6, HISP-NET maintains stable performance across all datasets, with optimal results concentrated within the $\lambda \in \{0.05, 0.1\}$ interval where both AUC and ACC reach their peaks. Beyond this range (*i.e.*, $\lambda \geq 0.2$), performance begins to de-

*Table 4.* Ablation study of the MinCut term ($\mathcal{L}_{mc}$) and Orthogonality term ($\mathcal{L}_{ortho}$) in the TAA regularizer on ABIDE.

| Method | AUC ↑ | ACC ↑ | SEN ↑ | SPE ↑ |
|---|---|---|---|---|
| w/o $\mathcal{L}_{mc}$&$\mathcal{L}_{ortho}$ | $77.11 \pm 1.55$ | $71.61 \pm 2.13$ | $70.91 \pm 4.20$ | $71.18 \pm 5.62$ |
| w/o $\mathcal{L}_{mc}$ | $75.47 \pm 1.04$ | $69.79 \pm 4.69$ | $67.01 \pm 9.98$ | $69.96 \pm 10.97$ |
| w/o $\mathcal{L}_{ortho}$ | $75.50 \pm 1.78$ | $69.51 \pm 2.89$ | $65.16 \pm 10.18$ | $72.01 \pm 10.78$ |
| HISP-NET | $\mathbf{78.05 \pm 1.55}$ | $\mathbf{72.90 \pm 1.32}$ | $\mathbf{72.73 \pm 4.75}$ | $\mathbf{74.22 \pm 4.95}$ |

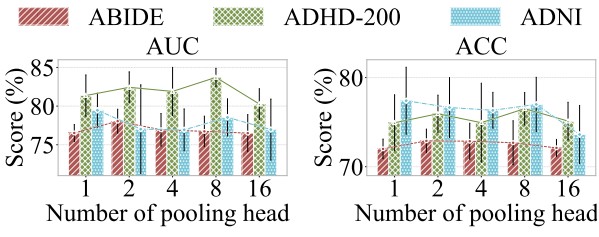

*Figure 5.* Impact of the number of pooling heads.

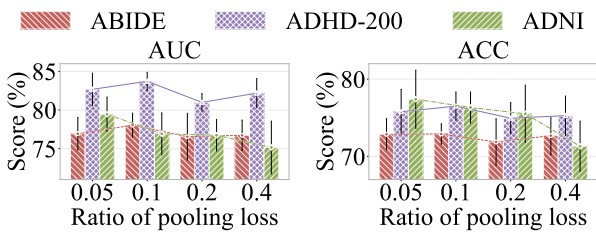

*Figure 6.* Impact of pooling loss ratio ($\lambda$).

grade, particularly on the ADNI dataset. This trend indicates that an excessive loss ratio can over-regularize the projection block, thereby suppressing the discriminative feature expression necessary for classification. The relatively stable plateau across the 0.05–0.1 range demonstrates that HISP-NET is robust to hyperparameter scaling.

**Impact of Residual Depth (Fig. 7).** The proposed HISP-NET generally achieves the best trade-off with a shallow residual configuration of 2 layers. Increasing the depth to 3 layers yields only marginal improvements on ABIDE, but leads to noticeable performance degradation on both ADHD-200 and ADNI. This trend demonstrates that a lightweight projection block is sufficient to capture discriminative parcellation features, circumventing the unnecessary optimization complexity and parameter redundancy associated with deeper architectures.

### A.2.5. DATASET-SPECIFIC HYPERPARAMETERS

To ensure reproducibility, Tab. 5 reports the optimal hyperparameter configurations. These specific values are determined through a grid search.

## B. Proof of Theorem 1

*Proof.* Let $\mathbf{C}^* = \sum_{i=1}^{N} \hat{\mathbf{c}}_i \mathbf{u}_i$ be the spectral decomposition of the ground-truth partition. Since $\mathbf{C}^*$ is a step function with sharp discontinuities (representing distinct functional

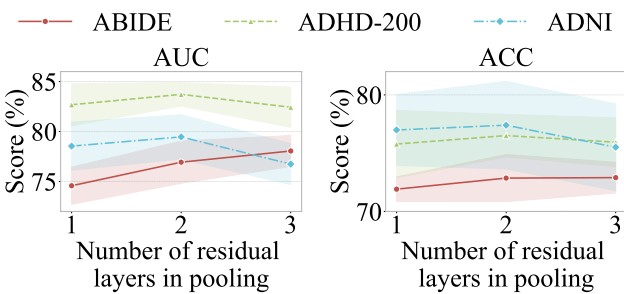

*Figure 7.* Impact of residual depth in the pooling module.

*Table 5.* Specific values of hyperparameters for the ABIDE, ADHD-200 and ADNI datasets.

| Hyperparameter | ABIDE | ADHD-200 | ADNI |
|---|---|---|---|
| dropout rate | 0.2 | 0.1 | 0.1 |
| pooling heads | 2 | 8 | 1 |
| residual depth | 3 | 2 | 2 |
| pooling loss ratio | 0.1 | 0.1 | 0.05 |

modules), graph signal processing principles imply significant energy in the high-frequency band ($\lambda_i \geq \tau$). Thus, the projection energy is strictly positive, as

$$\|\mathcal{P}_{\text{high}}(\mathbf{C}^*)\|_F = \left\| \sum_{\lambda_i \geq \tau} \hat{\mathbf{c}}_i \mathbf{u}_i \right\|_F = \epsilon > 0. \quad (17)$$

Standard encoding architectures (GNNs/GTs) inherently act as smoothing operators, asymptotically suppressing high-frequency components. Therefore, their effective hypothesis space is restricted to $\mathcal{H}_{\text{smooth}} = \{\mathbf{C} \mid \mathcal{P}_{\text{high}}(\mathbf{C}) = \mathbf{0}\}$. For any $\mathbf{C}_{model} \in \mathcal{H}_{\text{smooth}}$, there is $\|\mathcal{P}_{\text{high}}(\mathbf{C}_{model})\|_F = 0$.

By the orthogonality of the spectral basis, the approximation error is lower-bounded as

$$
\begin{aligned}
\|\mathbf{C}_{model} - \mathbf{C}^*\|_F^2 &= \|\mathcal{P}_{\text{low}}(\mathbf{C}_{model} - \mathbf{C}^*)\|_F^2 \\
&\quad + \|\mathcal{P}_{\text{high}}(\mathbf{C}_{model} - \mathbf{C}^*)\|_F^2 \\
&\geq \|\mathcal{P}_{\text{high}}(\mathbf{C}_{model}) - \mathcal{P}_{\text{high}}(\mathbf{C}^*)\|_F^2 \\
&= \|\mathbf{0} - \mathcal{P}_{\text{high}}(\mathbf{C}^*)\|_F^2 \\
&= \epsilon^2.
\end{aligned}
$$
$$(18)$$

Taking the square root yields $\|\mathbf{C}_{model} - \mathbf{C}^*\|_F \geq \epsilon$, proving that perfect approximation is asymptotically unreachable within $\mathcal{H}_{\text{smooth}}$. □

# C. Proof of Proposition 2

*Proof.* Let $\mathbf{L} = \mathbf{U}\mathbf{\Lambda}\mathbf{U}^\top$ be the normalized Laplacian with eigenvalues $0 = \lambda_1 < \lambda_2 \leq \cdots \leq \lambda_N$. The gradient norm corresponds to the Dirichlet Energy: $\|\nabla_{\mathcal{G}}\mathbf{Z}\|_F^2 = \mathrm{Tr}(\mathbf{Z}^\top \mathbf{L}\mathbf{Z})$.

Firstly, consider the linear dynamics of GNNs, formulated as $\mathbf{Z}^{(k)} = (\mathbf{I} - \alpha\mathbf{L})^k \mathbf{X}$ with $\alpha \in (0, 2/\lambda_N)$ ensuring stability. The gradient norm is

$$\|\nabla_{\mathcal{G}}\mathbf{Z}^{(k)}\|_F^2 = \sum_{i=1}^N \lambda_i (1 - \alpha\lambda_i)^{2k} \|\hat{\mathbf{x}}_i\|_2^2. \quad (19)$$

For a connected graph, non-trivial modes ($i \geq 2$) satisfy $\lambda_i > 0$. Under the stability condition, these modes satisfy strict contraction. Defining the rate $\rho = \max_{i \geq 2} |1 - \alpha\lambda_i| < 1$, it yields

$$\lim_{k \to \infty} \|\nabla_{\mathcal{G}}\mathbf{Z}^{(k)}\|_F = 0. \quad (20)$$

This confirms that GNNs structurally compress representations into the low-frequency eigenspace (smoothing).

Secondly, consider the global attention mechanism in Graph Transformers. Theoretical analysis on attention (Dong et al., 2021) proves that deep architectures undergo *rank collapse*, converging to a rank-1 consensus matrix $\mathbf{1}\mathbf{c}^\top$. Therefore, node discrepancies are eliminated, and the gradient norm vanishes, that is,

$$\lim_{l \to \infty} \|\nabla_{\mathcal{G}}\mathbf{Z}_{GT}^{(l)}\|_F = 0. \quad (21)$$

Thus, deep GTs suffer from *global homogenization*, failing to preserve high-frequency boundaries.

In contrast, HISP-NET employs a node-wise MLP $f_\theta$ (spectral projection), which is structurally decoupled from the Laplacian $\mathbf{L}$. By the Universal Approximation Theorem, $f_\theta$ can approximate the highest-frequency eigenvector $\mathbf{u}_N$ (corresponding to $\lambda_{\max}$) to arbitrary precision. Thus, the gradient norm is bounded only by the graph spectrum, *i.e.*,

$$\lim_{\mathbf{Z} \to \mathbf{u}_N} \|\nabla_{\mathcal{G}}\mathbf{Z}\|_F^2 = \mathbf{u}_N^\top \mathbf{L}\mathbf{u}_N = \lambda_{\max}, \quad (22)$$

Crucially, $\lambda_{\max}$ is a topological constant, independent of the model depth or architecture.

Therefore, comparing the asymptotic behaviors, we have

$$\lim_{k \to \infty} \|\nabla_{\mathcal{G}}\mathbf{Z}_{Base}\|_F = 0$$
$$\neq \sup_{\theta} \|\nabla_{\mathcal{G}}\mathbf{Z}_{\text{HISP}}\|_F = \sqrt{\lambda_{\max}}, \quad (23)$$

This verifies that HISP-NET retains the capacity to represent arbitrarily sharp boundaries between functional modules, overcoming the limitations of both GNNs and GTs. $\square$

