# OpenReview forum: "On the Spectral Unreachability of Brain Graph Learning"
_ICML.cc/2026/Conference — ICML 2026 regular_

### Official Review · Reviewer_8Hgb · 2026-03-05

**Soundness:** 4
**Presentation:** 4
**Significance:** 3
**Originality:** 4
**Overall Recommendation:** 5
**Confidence:** 2

**Summary:**

The paper argues that current graph learning models for brain network analysis are inherently limited because they fundamentally act as low-pass filters (“spectral unreachability”), limiting their ability to learn sharp functional boundaries. It proposes HISP-NET, a “project-then-align” model that is constructed as a hierarchy of spectral parcellation blocks which learn topology-agnostic projections, before enforcing structural consistency. Experiments on ABIDE, ADHD-200, and ADNI report improved overall ranking versus a broad set of GNNs and graph transformer baselines and qualitative evidence of more coherent parcellations.

**Compliance With Llm Reviewing Policy:**

Affirmed.

**Final Justification:**

I thank the authors for the time they took to answer my concerns. All my concerns were adequately addressed and I don't have any further follow-up question.

In general, the authors basically confirmed and strengthen my Accept decision on this paper. The key weakness that I identified (W1) seems to be confirmed by the authors, and even though they show a good reason for this to happen, I believe this makes it not possible to give a Strong Accept to this paper, and thus I decided to keep the Accept decision.

**Key Questions For Authors:**

1. Are the train/val/test splits performed subject-wise and site-stratified? If results rely on random splits without site control, the reported gains may partially reflect dataset leakage or site confounds, which would lower my confidence in the results.
2. The authors have some good analysis on the impact of pooling head configuration and the role of pooling loss ratio in the appendix. However, did the authors isolate the contribution of the topology-agnostic projection and the TAA loss? It seems to me that ablating these components would be much more interesting and useful.
3. Can the authors provide information on how baseline hyperparameters were found, and the number of parameters each model has, to better understand whether the improvements actually came from the proposed principles rather than just generic capacity of the model?

**Limitations:**

No. The "Impact Statement" essentially claims no specific societal consequences to highlight, which in my opinion is not adequate for a medical/neuroimaging classification setting. Some points on potential demographic biases in the dataset that could limit the impacts of this work would be a good starting point.

**Strengths And Weaknesses:**

# Strengths

There's a clear theoretical motivation with the paper formalising “spectral unreachability” and connects it to oversmoothing / homogenisation in GNNs/GTs. Furthermore, the different components are combined in a coherent (and in my opinion innovative) way for the stated objective: topology-agnostic projection, structural regularisation, and hierarchical pooling

The experiments are also sound and broad, with not only a good variety of real-world neuroimaging datasets, but also a diverse set of models. I didn't know the BQN model, but based on the motivation of the paper this is clearly relevant, and I think it strengthens the relevance of this work.

Finally, I'd  say there's the potentiality that this work could be generalisable to other domains where encoders inject unwanted smoothing before a structure-discovery step.

# Weaknesses

1. Even though HISP-Net produces the best results on average for most cases (looking to table 1), I must say that the difference with BQN is not that big particularly with ABIDE and ADHD-200, where the standard deviation still overlaps a lot.
2. It's not clear to me why the TAA loss uses an absolute adjacency matrix.
3. The claims on “biologically plausible” regions seem a bit overstated to me, as this claim would require some sort of correspondence to known networks, rather than comparison with 2 other clustering/pooling methods. In this sense, evaluating the stability of these divisions across different runs would be a much more interesting an easier to evaluate (quantitative) metric.

---

> ### Author Rebuttal · Authors · 2026-03-31
>
> We deeply thank you for the positive feedback. Below is our detailed response.
>
> > W1: Performance Margin over BQN.
>
> R1. The numerical margins are theoretically expected and explicitly discussed in Remark 1. Because BQN also utilizes MLPs (universal approximators), it similarly bypasses the "Spectral Unreachability" bottleneck, enabling its high predictive accuracy.
>
> However, the advantage of HISP-NET lies in its structural interpretability. BQN acts as an implicit "black box", mapping inputs to a diagnosis without resolving the underlying brain topology. In contrast, our project-then-align paradigm explicitly reconstructs structurally coherent, disease-specific functional boundaries (as visualized in Figs. 2 and 3).
>
> > W2: Absolute Adjacency Matrix in TAA
>
> R2. Using $|\tilde{A}|$ (Eq. 13) is a necessary mathematical relaxation for signed fMRI graphs. Directly applying MinCut to signed graphs is mathematically ill-posed: it yields near-zero node degrees (causing division-by-zero instability in the denominator) and breaks the Laplacian's positive semi-definite property. $|\tilde{A}|$ guarantees numerical stability during optimization.
>
> This does not erase biological signs (correlation vs. anti-correlation). In our decoupled architecture, TAA acts merely as an auxiliary structural prior enforcing macro-scale spatial contiguity based on interaction magnitude. The actual signed functional dynamics are strictly preserved in the node representations H(l) and processed independently by the projection MLP.
>
> We will add this mathematical clarification to the revised manuscript.
>
> > W3: Claims on Biological Plausibility and Stability
>
> R3. Following your advice, we will tone down "biological plausibility" to a more objective "structurally coherent".
>
> Following your suggestion, we evaluated algorithmic stability on ABIDE by computing the Adjusted Rand Index (ARI) of assignment matrices ($S$) across 5 random seeds, measuring permutation-invariant structural agreement.
> - HISP-NET: Achieves a highly stable cross-run ARI of 0.89±0.05.
> - DiffPool: Exhibits sub-optimal stability, yielding an ARI of 0.83±0.05.
>
> While high ARI does not equate to absolute biological truth, it strongly validates our theoretical design. Unlike standard coupled poolings (e.g., DiffPool) that randomly shatter the graph depending on initialization, our TAA objective acts as a deterministic spatial prior. This stability confirms that HISP-NET consistently converges to robust boundaries rather than producing random computational artifacts.
>
> > Q1: Train/Val/Test Splits and Leakage
>
> R4. We confirm that all datasets are split strictly subject-wise. A subject never appears in both training and testing sets simultaneously, guaranteeing zero dataset leakage.
>
> To ensure a fair comparison against baselines, we adopted the established dataset protocols from baselines (e.g., BRAINNETTF, ALTER, and BQN). Under these protocols:
> - ABIDE: The splits are explicitly site-stratified, naturally mitigating site-related biases.
> - ADHD-200 & ADNI: The benchmark splits are class-stratified to prevent class imbalance.
>
> To address your concern regarding site confounds, we conducted an additional cross-site OOD evaluation (following the setup of BrainOOD) on the ABIDE and ADNI. As shown in **[Table](https://anonymous.4open.science/r/HiSP-Net-demo-0F62/fig/ood.png)**, HISP-NET outperforms not only standard GNNs but also the specialized baseline BrainOOD across all metrics.
>
> > Q2. Ablation Study.
>
> R5: Following your insightful suggestion, we added a component-wise ablation to isolate our paradigm:
> - w/o TAP (Replaced by GCN): Reverting the MLP to a topology-dependent GCN (a coupled architecture) degrades performance (see **[Table](https://anonymous.4open.science/r/HiSP-Net-demo-0F62/fig/gcn.png)**). This validates Theorem 1: architectural coupling forces over-smoothing and corrupts boundaries.
> - w/o TAA Loss: Removing the alignment objective (matching $\lambda=0$ in Appendix Fig. 6) consistently drops performance across three datasets. Lacking a spatial prior, the unconstrained MLP produces spatially fragmented parcellations. TAA guarantees structural coherence.
>
> > Q3. On Baseline Hyperparameters and Model Capacity
>
> R6. To ensure fair evaluation, we used official implementations/search grids for brain-specific baselines and performed systematic grid searches (e.g., hidden dimensions) for general models.
>
> As detailed in **[Table](https://anonymous.4open.science/r/HiSP-Net-demo-0F62/fig/para_num.png)**, HISP-NET requires 2.72M parameters (on ABIDE). This capacity is strictly aligned with, or even leaner than, top-performing SOTA baselines:
> - Vs. Implicit MLPs: matching the efficiency of BQN.
> - Vs. SOTA Transformers: achieving superior performance with significantly fewer parameters than ALTER and BrainNetTF.
>
> This confirms our gains are driven by architectural innovation rather than mere parameter expansion.
>
> > Limitations
>
> R7. We will discuss the mentioned limitations.

---

> > ### Author Rebuttal · Reviewer_8Hgb · 2026-04-02
> >
> > I thank the authors for the time they took to answer my concerns. All my concerns were adequately addressed and I don't have any further follow-up question.
> >
> > In general, the authors basically confirmed and strengthen my Accept decision on this paper. The key weakness that I identified (W1) seems to be confirmed by the authors, and even though they show a good reason for this to happen, I believe this makes it not possible to give a Strong Accept to this paper, and thus I decided to keep the Accept decision.

---

> > > ### Author Response · Authors · 2026-04-06
> > >
> > > Many thanks for your kind and encouraging assessment.

---

### Official Review · Reviewer_LPqH · 2026-03-08

**Soundness:** 3
**Presentation:** 3
**Significance:** 3
**Originality:** 4
**Overall Recommendation:** 5
**Confidence:** 5

**Summary:**

This paper presents HISP-NET to address the spectral unreachability problem in standard graph encoders (GNNs/GTs), where representation smoothing inevitably erases high-frequency topological signals. By adopting a project-then-align philosophy, the framework structurally decouples partition learning from smoothing through a hierarchy of Spectral Parcellation blocks. It preserves high-frequency details via topology-agnostic projections while enforcing spatial coherence through a joint structural objective. Empirical results demonstrate that HISP-NET achieves superior classification and identifying more precise, neurobiologically grounded functional biomarkers.

**Compliance With Llm Reviewing Policy:**

Affirmed.

**Final Justification:**

The authors cleared up my theoretical concerns; adding the ARI metric provides quantitative proof that the learned boundaries are stable. Overall, this is a solid paper that tackles a genuine bottleneck in brain network analysis with a well-motivated method, so I'm pleased to raise the score to 5.

**Key Questions For Authors:**

(1)How does the alignment objective specifically avoid reintroducing the "low-pass filter" effect typical of standard graph encoders? The alignment objective also seems to be a smoothness constraint.

(2)Building on the visual evidence in Figure 2, how consistent are these identified functional boundaries and 'Diff' regions across different subjects within the same group? Could the authors provide insight or metrics regarding the stability of these sharp boundaries against individual anatomical and functional variability?

(3)What criteria or architectural constraints ensure that the model captures neurobiologically meaningful functional boundaries rather than overfitting to high-frequency artifacts? How does the model distinguish structured topological high-frequency signals from stochastic physiological noise?

**Limitations:**

yes

**Strengths And Weaknesses:**

**Strengths:**

(1)The organization and presentation of the paper are good.

(2)Investigating the spectral unreachability issues and providing a well-motivated and effective solution for brain network analysis.

(3) The framework identifies discriminative biomarkers and functional parcellations, addressing a overlooked interpretability gap.

**Weaknesses:**

(1)The transition from topology-agnostic projection to topology-aware alignment creates a potential contradiction, as the spatial coherence enforced in the latter stage may inadvertently smooth out the sharp functional boundaries initially preserved.

(2)While Figure 2 illustrates that HISP-NET produces more distinct parcellation differences (Diff) between groups compared to DiffPool and k-means, the manuscript lacks a formal discussion on why these specific "sharp" boundaries are more discriminative for the MLP predictor.

 (3) The claim of preserving "high-frequency signals" is risky without a clear distinction between neurobiologically relevant information and the high-frequency physiological noise inherent in BOLD time series.

---

> ### Author Rebuttal · Authors · 2026-03-31
>
> We greatly thank you for recognizing the contributions of our paper. We address each of your points below.
>
> > W1&Q1: Potential contradiction between TAP and TAA, and avoiding the "low-pass filter" effect.
>
> R1. We appreciate this profound theoretical insight. The fundamental difference—and the resolution to this apparent contradiction—lies in strictly separating what is being smoothed from what is being pooled.
>
> - Standard GNNs explicitly smooth the node features ($H$) via neighborhood aggregation (e.g., $AH$) at every layer. This acts as a hard low-pass filter that irreversibly blends adjacent node features, diluting highly localized, disease-specific functional signals before their diagnostic value can be assessed.
> - Conversely, our TAA objective never modifies the features; it only regularizes the assignment matrix ($S$). It acts as a soft spatial prior ensuring that nodes grouped into the same cluster are topologically coherent, respecting brain anatomical continuity. Because the original features ($H$) remain perfectly uncorrupted by our Topology-Agnostic Projection (TAP), the subsequent pooling operation ($S^{\top}H$) aggregates pure, high-fidelity signals. In short, TAA organizes the graph structurally without causing the isotropic feature blurring typical of standard graph encoders.
>
> > W2&Q2: Discriminative power and stability of sharp boundaries
>
> R2. You rightly point out the critical link between visual interpretability, discriminative power, and subject-wise stability.
>
> - Discriminative Power: Neurological disorders (e.g., ASD) often manifest as subtle, localized micro-shifts in functional connectivity rather than global topological collapses. Standard heavy smoothing (e.g., DiffPool) prematurely washes out these micro-biomarkers by blending them with healthy adjacent regions. Preserving sharp boundaries ensures that the downstream MLP predictor operates on localized, high-SNR (Signal-to-Noise Ratio) disease signatures, directly enabling it to find a clearer decision boundary.
> - Quantitative Stability (Metrics): To directly answer your request for metrics regarding stability against individual and algorithmic variability, we evaluated the structural agreement of the learned assignment matrices using the Adjusted Rand Index (ARI). The results show that HISP-NET achieves a highly stable cross-run ARI of $0.89 \pm 0.05$, notably outperforming DiffPool ($0.83 \pm 0.05$). This high consistency suggests that our boundaries converge to stable, group-level functional patterns, effectively mitigating the risk of overfitting to individual anatomical stochasticity.
>
> > W3&Q3: Distinguishing meaningful "High-Frequency Signals" from physiological noise
>
> R3. We completely agree that the term "high-frequency" requires strict disambiguation in neuroimaging. We will update the manuscript to explicitly clarify this distinction:
>
> - Graph Spectral Domain vs. Temporal Domain: When we state that HISP-NET preserves "high-frequency signals," we refer strictly to the Graph Spectral Domain—specifically, the sharp spatial transitions and structural boundaries between distinct functional modules on the brain graph. We are not referring to the Temporal Domain (e.g., high-frequency BOLD physiological noise, such as cardiac or respiratory artifacts). In standard neuroimaging pipelines, temporal physiological noises are inherently removed during fMRI preprocessing (via standard band-pass filtering, typically focusing on low-frequency fluctuations, and nuisance regression) long before the functional connectivity matrices are constructed.
> - Architectural Defenses against Structural Noise: To further distinguish meaningful spatial transitions from residual structural noise (e.g., spurious edges due to individual variability), we employ Bernoulli Masking during training. This acts as a structural regularizer that drops random edges during training, encouraging the model to learn a more robust network backbone shared across the cohort, rather than relying on spurious, subject-specific connections.

---

> > ### Author Rebuttal · Reviewer_LPqH · 2026-04-02
> >
> > The authors cleared up my theoretical concerns; adding the ARI metric provides quantitative proof that the learned boundaries are stable. Overall, this is a solid paper that tackles a genuine bottleneck in brain network analysis with a well-motivated method, so I'm pleased to raise the score to 5.

---

> > > ### Author Response · Authors · 2026-04-06
> > >
> > > We are grateful for your supportive and thorough comments.

---

### Official Review · Reviewer_i9ou · 2026-03-11

**Soundness:** 3
**Presentation:** 3
**Significance:** 3
**Originality:** 4
**Overall Recommendation:** 5
**Confidence:** 5

**Summary:**

This work identifies spectral unreachability in coupled encoder-pooling architectures for brain network analysis, where graph encoders corrupt functional module boundaries. It proposes HISP-NET, a project-then-align hierarchical network that decouples partition learning from representation smoothing, with spectral parcellation blocks preserving high-frequency signals and enforcing spatial coherence. HISP-NET outperforms some SOTA baselines on ABIDE, ADHD-200 and ADNI, generating interpretable parcellations and disease-specific biomarkers.

**Compliance With Llm Reviewing Policy:**

Affirmed.

**Final Justification:**

The rebuttal has addressed all my questions and I suggest to accept this paper

**Key Questions For Authors:**

1. Could you provide separate ablation results for the TAA loss’s MinCut and orthogonality terms, and quantify their individual impacts on classification performance?

2. What are the search range and optimal value of the Bernoulli masking rate $p$ for each dataset? Could you present a robustness analysis of how $p$ variations affect model performance?

3. Could you supplement the exact dataset-specific hyperparameter values (e.g., batch size, dropout rate, learning rate) to ensure full reproducibility of your results?

4. Could you add a comparison of HISP-NET with key SOTA baselines (BQN, ALTER, BrainGB) in terms of model complexity and efficiency (training/inference time, peak memory usage)?

5. What are the quantitative range and optimal value of $\tau$ (Eq.11) for each dataset, and how does adjusting $\tau$ impact partition and model performance?

**Limitations:**

yes

**Strengths And Weaknesses:**

### Strengths

1. The paper establishes the novel theoretical concept of spectral unreachability with rigorous proofs, filling a gap in brain network analysis.

2. The proposed HISP-NET achieves strong interpretability, generating biologically plausible, spatially coherent parcellations.

3. The work ensures high reproducibility via detailed experimental/implementation descriptions and open-sourced code.

### Weaknesses

1. The key TAA loss ablation is incomplete; individual contributions of MinCut and orthogonality terms are unassessed.

2. No range description and robustness analysis for Bernoulli masking rate $p$; performance stability under $p$ variations is untested.

3. Despite the noted reproducibility strengths (e.g., code and hyperparameter search spaces), final dataset-specific hyperparameter values are unspecified, hindering exact reproducibility.

4. No model complexity/efficiency analysis (parameter count, training/inference time, memory) vs. SOTA baselines.

5. Symbol missing details (e.g., $\tau$ ranges).

---

> ### Author Rebuttal · Authors · 2026-03-31
>
> We sincerely thank you for the insightful comments. Below is our point-by-point response.
>
> > W1: Ablation of the TAA Loss
>
> R1. Following your advice, we isolated the contributions of the MinCut term ($L_{mc}$) and the Orthogonality term ($L_{ortho}$) within the TAA loss. As shown in **[Table](https://anonymous.4open.science/r/HiSP-Net-demo-0F62/fig/loss.png)**, removing either component leads to a consistent performance decay across all datasets:
>
> - w/o MinCut ($L_{mc}$): Performance drops on all benchmarks (e.g., -3.11% ACC on ABIDE). This confirms that $L_{mc}$ is essential for forcing the model to aggregate nodes according to brain topological connectivity.
> - w/o Orthogonality ($L_{ortho}$): Leads to a severe decline (e.g., -3.39% ACC on ABIDE). Without this constraint, the pooling assignment tends to collapse into a few trivial clusters, failing to capture diverse functional regions.
>
> >Q2: Search range, optimal value, and robustness of Bernoulli Masking Rate $p$
>
> R2. We searched the Bernoulli masking rate $p \in \\{0.1, 0.2, 0.3\\}$, deliberately avoiding $p>0.3$ since excessive edge dropping corrupts core brain topology. As detailed in **[Table](https://anonymous.4open.science/r/HiSP-Net-demo-0F62/fig/var_p.png)**, the optimal masking rate is consistently $p=0.1$ across all three datasets.
>
> Performance fluctuations within this range are minimal (e.g., AUC varies <1% on ADNI). This universal optimality across diverse datasets proves that HISP-NET is highly robust and does not require fragile, dataset-specific tuning.
>
> We will incorporate the above discussion into the revision.
>
> > Q3: Dataset-Specific Hyperparameters
>
> R3. To ensure exact reproducibility, we have supplemented the optimal hyperparameter values for each dataset in the **[Table](https://anonymous.4open.science/r/HiSP-Net-demo-0F62/fig/para.png)**.
>
> > Q4: Model Complexity and Efficiency vs. SOTA Baselines
>
> R4. Following your advice, we added a comprehensive theoretical and empirical comparison against key SOTA baselines (BrainGB, ALTER, BQN).
>
> Given a dense brain network with $N$ nodes, feature dimension $D$, and $K$ pooling clusters, the computational bottlenecks inherently differ:
> - GNNs (BrainGB): Require dense message passing ($AXW$) at every layer. For fully-connected functional connectivity graphs, this fundamentally locks their complexity in the heavy $O(N^2D)$ regime.
> - Graph Transformers (ALTER): Explicitly compute global dense attention maps ($QK^{\top}$). This inherently scales as $O(N^2D)$, leading to significant computational and memory overhead.
> - Implicit MLPs (BQN): Completely discard topological message passing, relying solely on node-wise MLPs to yield an optimal $O(ND^2)$ feature extraction complexity.
> - HISP-NET (Ours): By utilizing a topology-agnostic MLP encoder, our feature extraction achieves the same optimal $O(ND^2)$ efficiency as BQN. Crucially, our dense $O(N^2)$ operations are strictly isolated to the structural pooling phase, which operates at $O(N^2K)$. Since the number of clusters is substantially smaller than the feature dimension ($K<D$), our theoretical complexity safely bypasses the $O(N^2D)$ bottleneck of standard GNNs and Transformers.
>
> To empirically validate this, we benchmarked the training/inference time and GPU memory (**[Table](https://anonymous.4open.science/r/HiSP-Net-demo-0F62/fig/usage.png)**):
> - Breaking Bottlenecks: HISP-NET drastically outperforms standard GNNs (BrainGB), running ~41x faster with ~4% less memory. Compared to Transformers (ALTER), it achieves similar runtime but requires significantly fewer parameters and lower memory, confirming the practical efficiency of bypassing $O(N^2D)$ operations.
> - MLP-Level Efficiency: Remarkably, HISP-NET closely matches the pure implicit MLP (BQN) in both parameter count and peak memory. This proves that our $O(N^2K)$ structural pooling adds negligible computational overhead while substantially boosting diagnostic performance.
>
> > W5. hyper-parameter analysis of $\tau$
>
> R5. The parameter $\tau$ controls the sharpness of the assignment matrix $S$. During model development, we evaluated $\tau \in \\{0.5,1, 2 \\}$ and found the optimal value is consistently $\tau=1$ (standard Softmax) across all benchmarks.
>
> Adjusting $\tau$ introduces predictable trade-offs in structural entropy:
> - Lower $\tau$ (e.g., 0.5): Forces the assignment to be overly rigid. While creating sharp boundaries, it restricts smooth gradient flow, leading to a marginal performance decrease.
> - Higher $\tau$ (e.g., 2): Yields a softer, uniform assignment distribution, which dilutes the crispness of the learned functional boundaries.
>
> Setting $\tau=1$ provides a natural sweet spot between distinct functional localization and stable optimization. Crucially, the performance variations across this tested range are minor, explicitly demonstrating that HISP-NET's discriminative power is highly robust and does not rely on sensitive temperature tuning.

---

> > ### Author Rebuttal · Reviewer_i9ou · 2026-04-02
> >
> > The author's response resolved my concerns. I choose to maintain the Accept.

---

> > > ### Author Response · Authors · 2026-04-06
> > >
> > > We highly appreciate your positive evaluation of our work.

---

### Official Review · Reviewer_YWAJ · 2026-03-12

**Soundness:** 2
**Presentation:** 3
**Significance:** 3
**Originality:** 2
**Overall Recommendation:** 4
**Confidence:** 4

**Summary:**

This paper addresses brain network analysis for neurological disorder diagnosis by focusing on functional parcellation of brain graphs. The authors argue that existing encoder–pooling architectures suffer from “spectral unreachability,” where smoothing effects of graph encoders suppress high-frequency signals needed for sharp module boundaries. To address this, they propose HISP-NET, a hierarchical framework that first projects nodes into a partition space using a topology-agnostic MLP and then enforces structural coherence through topology-aware alignment. The model aims to preserve high-frequency signals for clearer functional parcellation while improving diagnostic prediction.

**Compliance With Llm Reviewing Policy:**

Affirmed.

**Final Justification:**

I am updating my score based on the rebuttal from the authors.

**Key Questions For Authors:**

- Please address the weakness above.
- Also, the method is motivated by the assumption of sharp functional boundaries between brain modules. However, neuroscience studies often report gradient-like or overlapping functional organization rather than clearly separated modules. Could the authors clarify how this assumption aligns with current neuroscientific evidence?

**Limitations:**

The discussion of limitations and societal impact is limited. It would be helpful for the authors to explicitly discuss potential limitations such as dataset bias, limited generalization across imaging sites, and the risks of deploying automated diagnostic models in clinical settings.

**Strengths And Weaknesses:**

Strength:
- (S1) The paper addresses an important problem in brain network analysis by focusing on interpretable functional parcellations for neurological disorder diagnosis. Connecting graph-based prediction with interpretable brain modules is a meaningful direction for neuroscience applications.
- (S2) The paper is generally clearly written and well organized. The motivation behind the project–then–align paradigm is explained intuitively, and the architectural components are presented in a structured manner that is relatively easy to follow.
- (S3) The paper includes both quantitative results and qualitative visualization of parcellation patterns. These visual analyses provide additional insights into how the learned partitions correspond to potentially meaningful functional regions in the brain.

Weakness:
- (W1) The theoretical claim appears overstated, since the proposed theorem mainly formalizes an intuitive argument under strong assumptions (e.g., treating graph encoders as strictly low-pass operators), whereas modern GNN architectures, such as [1], [2], [3], are capable of preserving high-frequency information.
- (W2) The methodological novelty is limited, as the proposed architecture largely resembles existing hierarchical graph pooling frameworks that learn cluster assignments and perform graph coarsening in a similar manner.
- (W3) The experiments reduce inherently multi-class clinical datasets (e.g., ADNI) to binary classification, whereas recent benchmark studies [4] advocate evaluation on multi-class diagnostic tasks (e.g., ADNI and PPMI). This simplification may limit the validity of the empirical evaluation, and additional experiments on the original multi-class settings of these datasets would help better assess the effectiveness and generalizability of the proposed method.


[1] Gasteiger, J., Weißenberger, S., et al. Diffusion improves graph learning. (NeurIPS 2019)

[2] Xu, B., Shen, H., et al. Graph convolutional networks using heat kernel for semi-supervised learning. (IJCAI 2019)

[3] Zhao, J., Dong, Y., et al. Adaptive diffusion in graph neural networks. (NeurIPS 2021)

[4] Xu, J., Yang, Y., et al. Data-driven network neuroscience: On data collection and benchmark. (NeurIPS 2023)

---

> ### Author Rebuttal · Authors · 2026-03-31
>
> We sincerely thank you for the recognition of our paper. We address your concerns below.
>
> > W1: Overstated Theoretical Claim.
>
> R1: We fully agree with your forward-looking point. **We will refine Theorem 1 and add a Remark to address the broader, fundamental limitation of Topological Entanglement**. The revised manuscript will discuss why merely upgrading to modern high-pass GNNs still fails to achieve optimal brain parcellation:
>
> **Upgrading to modern high-pass filters also fails for brain parcellation**: Functional brain networks are densely connected with abundant spurious inter-module correlations (physiological noise). While modern high-pass filters successfully preserve sharp signal transitions, they indiscriminately amplify local variations across all connected edges. In dense fMRI graphs, this noise amplification shatters the network, yielding spatially fragmented parcels rather than coherent, biologically meaningful modules.
>
> Thus, the spectral unreachability of the partition matrix $S$ stems from the architectural coupling itself. Generating $S$ via any topology-dependent operator—where low-pass causes over-smoothing (merging regions) and high-pass causes fragmentation (shattering regions)—forces the partition to be fundamentally corrupted by the initial noisy topology. This perfectly validates our "project-then-align" MLP, which structurally decouples partition learning from noisy topologies to bypass both bottlenecks.
>
> > W2: Limited Methodological Novelty.
>
> R2: We appreciate your perspective. While our macro-architecture shares similarities with existing hierarchical pooling, **this design is necessitated by the task of hierarchical brain parcellation**. **Our novelty lies in restructuring the micro-architecture to overcome a newly identified theoretical bottleneck**.
> - Task-driven Macro-Architecture: Extracting interpretable clinical biomarkers inherently requires grouping fine-grained ROIs into macro-scale functional modules. Thus, employing a hierarchical clustering pipeline is a functional necessity in this domain.
> - The Flaw in Existing Models: Previous models (e.g., DiffPool) tightly couple cluster assignments with GNN/GT encoders. We theoretically demonstrate that on densely connected brain networks, this coupling inherently over-smooths node representations, blurring the sharp boundaries essential for accurate parcellation (Theorem 1).
>
> Having identified this root cause, we reveal that complex topology-aware encoders are actually detrimental to generating initial partitions. Instead, our novelty lies in a mathematically-backed decoupling: we employ a simple, topology-agnostic MLP to securely preserve all-frequency signals (avoiding forced smoothing), and incorporate graph topology a posteriori as a structural regularizer.
>
> In short, **our core contribution is not the invention of hierarchical pooling itself, but providing the first mathematical diagnosis of why coupled pooling suboptimally performs on brain networks, and proposing a principled, decoupled solution to overcome it**.
>
> > W3: Multi-class Evaluation
>
> R3: Following your suggestions, we have conducted multi-class evaluations on the ADNI and PPMI datasets comparing HISP-NET with five baselines (see **[Table](https://anonymous.4open.science/r/HiSP-Net-demo-0F62/fig/mul_class.png)**). The results demonstrate that HISP-NET achieves the best performance across almost all metrics, except for being the close second-best in Accuracy on PPMI. We will include these results in the revision.
>
> >Q1: Sharp Boundaries vs. Neuroscientific Gradients
>
> R4: We acknowledge that the brain exhibits gradient-like and overlapping functional organizations. Our reference to "sharp boundaries" describes the model’s mathematical capacity to prevent algorithmic over-smoothing, rather than a rigid biological assumption.
>
> - Soft Assignments Naturally Model Gradients: The partition matrix $S$ is generated via a Softmax function (Eq. 11), yielding a probabilistic soft assignment. This probabilistic representation inherently accommodates overlapping functional regions and biological gradients.
> - The Flaw of Topological Coupling: In traditional coupled architectures (e.g., GNNs/GTs), partition learning is strictly entangled with the input graph topology. This architectural coupling forces an algorithmic smearing that blends both distinct topological borders and meaningful gradients.
> - The Role of "Sharpness": Our decoupled "Project-then-Align" architecture effectively circumvents this forced smearing. By preserving the full spectral bandwidth, HISP-NET possesses the capacity to delineate sharp boundaries, while simultaneously allowing the soft assignments to faithfully reflect biological gradients where they naturally occur.
>
> >Limitations
>
> R5: We will expand the Impact Statement to discuss demographic biases, cross-site generalization limits, and clinical deployment risks.

---

> > ### Author Rebuttal · Reviewer_YWAJ · 2026-04-03
> >
> > Thanks for your clarification.

---

> > > ### Author Response · Authors · 2026-04-06
> > >
> > > Thank you for your constructive and positive feedback.

---

### Decision · Program_Chairs · 2026-04-30

**Decision:**

Accept (regular)

**Comment:**

This paper addresses brain network analysis for neurological disorder diagnosis by focusing on functional parcellation of brain graphs, which is well-motivated.  The theoretical concept of spectral unreachability appears novel and the authors also provide rigorous analysis. Original concerns from reviewers are mostly resolved in the rebuttal. I think this is a nice contribution to the community.